# Diagnosing and exploiting the computational demands of videos games for deep reinforcement learning

## Abstract

Humans learn by interacting with their environments and perceiving the outcomes of their actions. A landmark in artificial intelligence has been the development of deep reinforcement learning (dRL) algorithms capable of doing the same in video games, on par with or better than humans. However, it remains unclear whether the successes of dRL models reflect advances in visual representation learning, the effectiveness of reinforcement learning algorithms at discovering better policies, or both. To address this question, we introduce the Learning Challenge Diagnosticator (LCD), a tool that separately measures the perceptual and reinforcement learning demands of a task. We use LCD to discover a novel taxonomy of challenges in the *Procgen* benchmark, and demonstrate that these predictions are both highly reliable and can instruct algorithmic development. More broadly, the LCD reveals multiple failure cases that can occur when optimizing dRL algorithms over entire video game benchmarks like *Procgen*, and provides a pathway towards more efficient progress.

## 1 Introduction

Gibson famously argued that "The function of vision is not to solve the inverse problem and reconstruct a veridical description of the physical world. [... It] is to keep perceivers in contact with behaviorally relevant properties of the world they inhabit" (reviewed in Warren 2021). The field of deep reinforcement learning (dRL) has followed Gibson's tenet since the seminal introduction of deep Q-networks (DQN) (Mnih et al., 2015). DQNs rely on reward feedback to train their policies and perceptual systems at the same time to learn to play games from a *tabula rasa*. This end-to-end approach of training on individual environments and tasks has supported steady progress in the field of dRL, and newer reinforcement learning algorithms have yielded agents that achieve human or super-human performance in a variety of challenges – from Chess to Go and from Atari games to Starcraft (Mnih et al., 2015; Silver et al., 2017; 2018; Vinyals et al., 2019). But Gibson also argued that the ecological niche of animals allows them to exploit task-agnostic mechanisms to simplify the perceptual or behavioral demands of important tasks, like how humans rely on optic flow for navigation (Warren, 2021). In the decades since Gibson's writings, it has been found that humans can efficiently find or learn bespoke perceptual features that aid performance on a single task (Li et al., 2004; Scott et al., 2007; Roelfsema et al., 2010; Emberson, 2017), or they can exploit previously learned generalist representations and task abstractions that are useful across multiple tasks and environments (Wiesel & Hubel, 1963; Watanabe et al., 2001; Emberson, 2017; Lehnert et al., 2020; O'Reilly, 2001). While there have been attempts at building similarly flexible dRL agents through meta-reinforcement learning (Frans et al., 2018; Xu et al., 2018; 2020; Houthooft et al., 2018; Gupta et al., 2018; Chelu et al., 2020; Pong et al., 2021), these approaches ignore the complexities of perceptual learning and carry large computational burdens that limit them to simplistic scenarios. There is a pressing need for approaches to training dRL agents that can meet the computational demands of a wide variety of environments and tasks.

One way to build generalist agents is to first reliably diagnose where the computational challenges of a given environment and task lie and adjust the agent to those demands. Is the perceptual challenge onerous? Is the reward signal for credit assignment especially sparse? Even partial answers to these questions are instructive for improving an agent, for instance, by pre-determining the extent to which

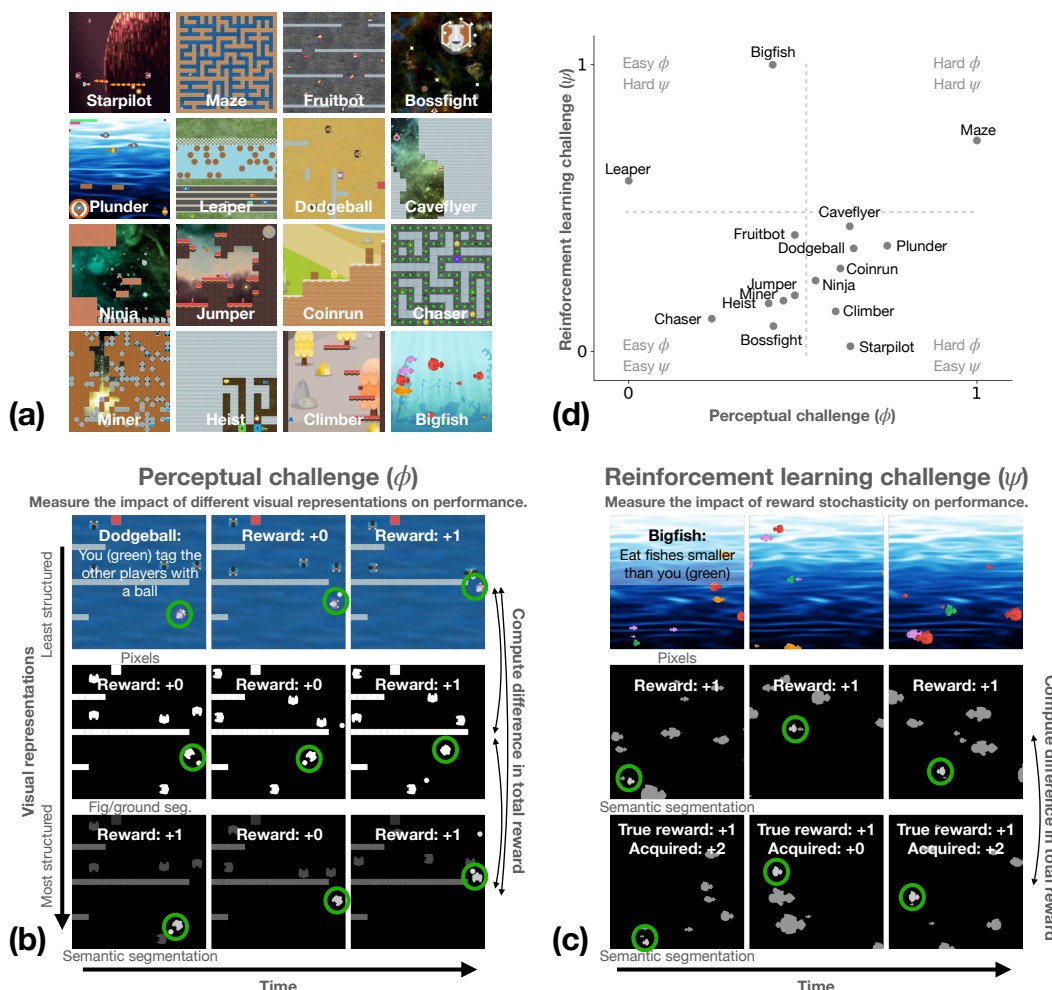

Figure 1: **The Learning Challenge Diagnosticator (LCD) separately measures the reinforcement learning or the perceptual challenge of an environment and task.** (a) We use LCD to compute perceptual and reinforcement learning challenges in each game of the *Procgen* Benchmark. (b) Perceptual challenges are assessed by comparing the total reward accrued by agents that learn over visual representations with different amounts of structure: pixels, figure/ground segmentation masks, and semantic segmentation masks (see Appendix. A.6). (c) Reinforcement learning challenges are computed by feeding agents perceptually organized scenes (i.e., semantically segmented; second and third rows), and then comparing the total reward accrued by one agent playing in an environment with stochastic manipulations of reward (randomly masked and modulated; third row) to another agent playing in the normal environment (second row). (d) The taxonomy of *Procgen* reveals each game's relative perceptual and reinforcement learning challenge.

it relies on feedback from the world to tune its policy and perception versus drawing from previously learned representations and task abstractions. The introduction of diverse video game challenges for dRL, such as the *Procgen* Benchmark (Cobbe et al., 2020), can serve as a starting point for this investigation. For example, take the game "Plunder" from *Procgen* (Figure 1a). Plunder has simple gameplay rules but poses a visual challenge: an agent is asked to shoot all objects that look like a provided cue. The difficulty here lies in assessing whether each object in the environment is the same or different than the cue; a visual routine that is difficult to learn for neural networks (Vaishnav et al., 2022; Kim et al., 2018). For this reason, an agent that can draw from prior experience in learning relevant perceptual routines may perform better than one with a perceptual system tuned for this specific task from scratch. In contrast, the objects and environments of a game like "Leaper"

(Figure 1a) can be identified by color. The game asks agents to use those objects to get from one side of the screen to the other, a task that can likely be learned from reward feedback.

**Contributions.** The ability to systematically, autonomously, and reliably identify an appropriate strategy for learning a task in a given environment would enable the design of generalist dRL agents that can flexibly adapt to challenges as humans do. To make progress towards this goal, we introduce the Learning Challenge Diagnosticator (LCD), a novel tool that identifies the specific computational challenges of an environment and task. When applied to video games, like those in *Procgen*, the LCD measures the difficulty of encoding task-relevant perceptual properties versus the difficulty of optimal policy discovery (i.e., determining which actions lead to rewarding outcomes).

- We develop a modified, parameterizable version of *Procgen*, to systematically manipulate the perceptual and reinforcement learning challenges presented to agents in each individual game. We provide our version of *Procgen* and all experimental code at `https://anonymous.4open.science/r/lcd-procgen/`.

- The LCD reveals a novel taxonomy of computational challenges in the games of our modified *Procgen*. Some are more visually complex, some are more challenging from the point of view of reinforcement learning, and others strain agents across both of these axes. This taxonomy is preserved across different dRL algorithms and the heterogeneity of the challenges associated with the *Procgen* games suggests that adopting a single "one size fits all" approach to learning, as is standard in dRL, is suboptimal.

- To address the visual challenges of *Procgen* games, we adopt a self-supervised visual front-end, which learns perceptual groups from motion cues in games without reward feedback. To test if reinforcement learning challenges can, in principle, be alleviated, we develop a proof-of-concept approach with reward shaping (Skinner, 1938).

- We exploit the computational taxonomy of *Procgen* revealed by the LCD to shape the design of agents for each game. These agents learn more efficiently and perform significantly better than one-size-fits-all dRL agents, suggesting the potential for "adaptive" agents.

## 2 RELATED WORK

**Representation learning in dRL** Perhaps the main contribution of DQNs was to demonstrate that it is possible to jointly learn visual representations and policies through reward maximization from a *tabula rasa* to achieve high scores on Atari games (Mnih et al., 2015). Despite this extraordinary achievement, the extrinsic rewards available to algorithms like DQNs have been found to yield poor sample complexity, limit model scale, and cap maximum performance (Jaderberg et al., 2016; Laskin et al., 2020). For this reason, there has been a growing number of accounts showing that standard reward learning can be augmented with auxiliary learning objectives and/or pre-training via self-supervision to take steps towards ameliorating limitations of reward-based learning (Jaderberg et al., 2016; Shelhamer et al., 2017; Stooke et al., 2021; Laskin et al., 2020; Dittadi et al., 2021; Zhang et al., 2020; Radosavovic et al.; Xiao et al.). Relatedly, it has been shown that initializing agents with structured visual representations can lead to faster learning and better performance in reinforcement learning (Tassa et al., 2018; Davidson & Lake, 2020; Laskin et al., 2020; Stooke et al., 2021).

**Measuring visual challenges in artificial intelligence** An essential contribution that the cognitive sciences have made to artificial intelligence over the past decade has been the introduction of computational challenges that are easy for humans to solve but strain the capabilities of neural networks. These challenges have identified deficiencies in the solutions deep neural networks (DNNs) tend to learn for contour tracing (Linsley et al., 2018b; Kim et al., 2020; Tay et al., 2021), segmentation (Kim et al., 2020), object tracking (Linsley et al., 2021), object recognition (Geirhos et al., 2021; 2018), and visual reasoning problems (Fleuret et al., 2011; Kim et al., 2018; Vaishnav et al., 2022). The essential lesson from these studies is that measuring model performance on data sampled from the same distribution that training data came from is not sufficient for establishing challenges and measuring progress on them (Funke et al., 2021). Instead, it is critical to test models on out-of-distribution samples to find and evaluate challenges (Geirhos et al., 2021; 2018; Linsley et al., 2021; 2020). We adopt this strategy to measure dRL agent performance in this work.

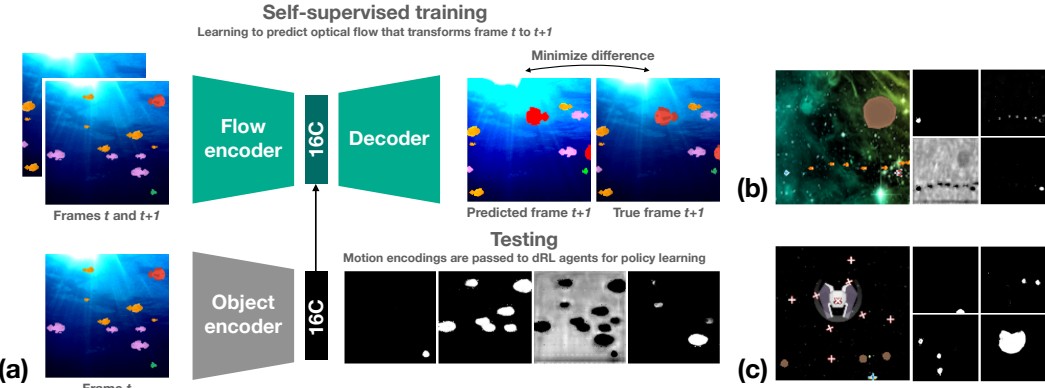

Figure 2: **A DNN that is self-supervised to predict future frames also learns a concomitant representation of object-ness that is useful for reinforcement learning.** (**a**) Our routine for self-supervising perceptual representations that can solve challenges identified by the LCD utilizes three networks: a ResNet18 (He et al., 2015) for predicting the optic flow between frame $t$ to frame $t+1$ (*Flow encoder*), a ResNet18 feature pyramid network (FPN) with a 16-channel recurrent neural network (Linsley et al., 2018a) to encode the content of frame $t$ (*Object encoder*), and an FPN-ResNet18 decoder that receives the two encoder's outputs to transform frame $t$ to frame $t+1$ via a differentiable warping module. After optimizing for next-frame prediction, the *Object encoder* learns object-like representations that can be used for reinforcement learning without additional training. This approach generates reasonable segmentations of the important objects in *Procgen* games, including Bigfish, Starpilot (**b**), and Bossfight (**c**).

**Biologically inspired mechanisms for improving neural networks** There are multiple examples of the DNN challenges being resolved by the introduction of biologically inspired mechanisms. Mechanisms for attention have made significant contributions by enabling DNNs to limit noise and clutter, amplify task-relevant features, and subsequently achieve better performance and sample efficiency during learning (Linsley et al., 2019). Others have found that horizontal connections, inspired by those in the early visual cortex of primates, can resolve DNN limitations in solving contour tracing tasks (Linsley et al., 2018a). There is also evidence that forcing dRL agents to acquire compositional abstractions, inspired by those described in the cognitive sciences, helps them learn policies that generalize to novel environments (Lehnert et al., 2020). The mechanisms we propose to solve the computational challenges revealed by the LCD in *Procgen* are similarly inspired by solutions to those problems adopted by primates and humans.

**Metrics for measuring progress in dRL** Progress in dRL has been classically measured by the aggregate rewards and sample efficiency of models in games (Taylor & Stone, 2009; Agarwal et al., 2021; Kirk et al., 2021). The *Procgen* benchmark took a major step forward beyond what is standard in the field by measuring agent performance on game parameterizations that fell outside of the distribution used for training. While current metrics can adjudicate between different algorithms based on performance in a game, they do not offer insights into the specific computational challenges that agents face. Our LCD therefore represents a major conceptual leap beyond current approaches to measuring progress in dRL, by quantifying the specific perceptual and reinforcement learning challenges found in a given game, and enabling dRL researchers to more precisely resolve the deficiencies of agents.

## 3   THE LEARNING CHALLENGE DIAGNOSTICATOR

The ability to identify and measure the computational challenges a dRL agent faces is an important step toward designing better algorithms for reinforcement learning. However, to the best of our knowledge, this problem has not yet been addressed in the field. The LCD achieves this goal through a three-step procedure, which reveals the computational challenges facing an agent. By revealing the computational demands of every game, the LCD enables more efficient algorithm development and insights into how these demands may interact – or not – during learning (see Section 4).

Here, we focus on identifying the perceptual versus reinforcement learning challenges offered by any specific game, yielding the following steps: (*i*) We begin by modifying a game to put its perceptual representations and reinforcement learning problem under experimental control. (*ii*) Next, a set of agents learn to play different versions of a game. In each version, either the perceptual challenge is perturbed while the reinforcement learning challenge is held constant, or vice versa. These perturbations are implemented by sampling either a different perceptual representation of the game or reward scheme. (*iii*) Finally, we measure the integrated reward trajectories of agents in response to each type of perturbation. By separately averaging the scores of agents facing perceptual perturbations and reward perturbations, we compute scores describing the perceptual challenge ($\phi$) and the reinforcement learning challenge ($\psi$) of each game (Fig. 1b and c).

**Procgen benchmark.** We applied the LCD to *Procgen* (Cobbe et al., 2020), a challenging dRL benchmark consisting of 16 games, each with distinct graphics and gameplay. *Procgen* is open-source and implemented in Python and C++, which made it possible to customize a variety of game parameters for implementing the LCD. The parameters we manipulated were game assets (sprites and backgrounds) and reward functions.

*Procgen* generates game levels procedurally, enabling precise control over the levels used for training versus testing and as a result, proper tests of generalization. We followed the training and testing protocol outlined in (Cobbe et al., 2020) in each of our experiments. In brief, all agents were trained for 200M steps across 500 training levels and evaluated on *Procgen's* held-out games to measure generalization.

**dRL algorithms.** We apply the LCD to agents trained with either the standard and popular proximal policy optimization (Schulman et al., 2017) (PPO) or phasic policy gradient (PPG) (Cobbe et al., 2021), a more recent algorithm that attained state-of-the-art performance in *Procgen*. PPO hyperparameters were selected to match those used in (Cobbe et al., 2020) for *Procgen's* hard mode. Similarly, for PPG we used the hyperparameters from (Cobbe et al., 2021). The walltime for a single training run of one of these agents averaged 24 hours per game on a standard 16-core Intel Xeon Gold 6242 CPU with a 24G Titan RTX GPU. We ran $\sim 450$ experiments across 32 GPUs, which took over $10K$ GPU hours of computing time. Agent performance at test time was computed using the interquartile mean (IQM) of the final returns (Agarwal et al., 2021).

**LCD: Perceptual challenge.** To perturb the perceptual complexity in a game and measure its perceptual challenge, we varied the visual input provided to an agent when learning to play the game. Motivated by decades of cognitive science work on the visual representations that humans rely on for efficiently learning to interact with their environments (Ullman, 1984; Roelfsema et al., 2000; Roelfsema, 2006b), we tested how well agents could learn to play games when given three different types of visual inputs with varying structure: the original frame, figure-ground segmentations of each frame, or semantic segmentations of each frame (Fig. 1b,c). Figure-ground segmentations labeled background and object pixels differently, whereas semantic segmentations labeled each element of the frame differently, with consistent labels across frames (see Appendix. A.6).

We measured the perceptual challenge of each game by comparing the performance of three agents trained to solve it, where each agent learned a policy over one of the three different types of visual inputs (Appendix Fig. 6). In this condition, rewards and the reinforcement learning challenge in games were kept as normal. We began by computing performance for each agent as the area under the cumulative reward curves (AUC) achieved by agents during generalization (Appendix Fig. 8), which implicitly captured both maximum performance and the sample efficiency needed to get there. Next, we computed the relative change in AUC from the original frame to figure-ground and semantic AUCs, and took the average of these ratios as our final score $\phi$ (Fig. 1b). The perceptual challenge score $\phi$ of each game was then normalized to the range $[0, 1]$ according to the min and max $\phi$ across all games (Appendix A.2). A game with a $\phi$ approaching 1 indicates that it presents a greater perceptual challenge for reward-based learning, whereas a small $\phi$ means that reward-based discovery of task-relevant features is sufficient to perform well on that task.

**LCD: Reinforcement learning challenge.** To perturb reinforcement learning in a game, we manipulated the sparsity of rewards available to an agent. By manipulating reward sparsity, we were able to degrade "learning signal" available to an agent while also preserving the optimal policy,

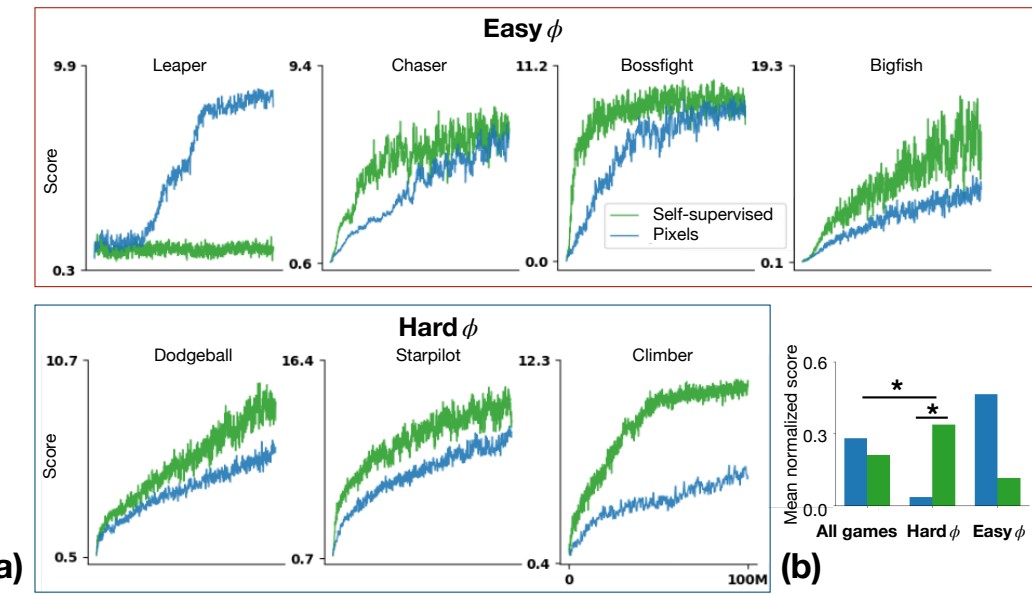

Figure 3: **Self-supervised pre-training for learning object-ness addresses perceptual challenges in *Procgen* identified by the LCD.** (**a**) We measured the impact of our self-supervised perceptual front-end for learning games classified as having perceptual challenges (Hard $\phi$) or not (Easy $\phi$). The game which is helped the least by the front-end, "Leaper", requires agents to recognize colors – a feature that is not preserved in the front-end. (**b**) When looking at aggregate performance across the entirety of *Procgen*, the benefit of pre-training is only seen on those games that the LCD deems as perceptually challenging (Hard $\phi$), indicating that the current state of progress in dRL is being slowed by an inability to find benchmarks (or subsets of benchmarks) which are aligned with the algorithmic solutions being tested. One-tailed $t-$tests assessing the effect of learning from the perceptual front-end versus pixels are denoted by lines, $* = p < 0.05$.

across all perturbations (Appendix A.3). We did this by decreasing the probability $p$ of receiving a reward at every rewarding event, while also scaling reward magnitudes by $1/p$ to preserve the expected magnitude of rewards. We used reward perturbations of $p \in \{1, 0.75, 0.50, 0.25\}$ (Appendix Fig. 7). All agents were given semantic segmentations as inputs, which minimized the perceptual challenge of games and controlled for potential perceptual confounds, such as how the perceptual complexity of the pixel inputs of a given game may non-linearly interact with this reinforcement learning perturbation.

We measured the reinforcement learning challenge of each game by comparing the performance of agents trained to solve each of the reward perturbations (Appendix Fig. 9). First, we again computed AUCs of the generalization reward curves for each agent, as above, but here for different values of $p$. Next, we computed the absolute AUC change from an agent trained on one $p$ to the next and averaged across all changes, yielding the score $\psi$ (Appendix A.4), which was normalized to $[0, 1]$ in the same way as $\phi$. A $\psi$ close to 1 means that it presents a significant reinforcement learning challenge. Conversely, a game with a small $\psi$ indicates that reward sparsity is not the primary challenge.

**Diagnosing the computational challenges of the Procgen benchmark.** We began by applying the LCD to PPO agents trained to solve each game in *Procgen*. Doing so revealed a clear taxonomy of the computational challenges of each game (Fig. 1d): Fruitbot, Jumper, Miner, Heist, Chaser, and Bossfight are, in relative terms, perceptually easy and have easy reinforcement learning. Caveflyer, Dodgeball, Plunder, Coinrun, Ninja, Climber, and Starpilot are perceptually hard and have easy reinforcement learning. Bigfish and Leaper are perceptually easy and have hard reinforcement learning, and Maze is challenging for perception and reinforcement learning. The challenges of Maze, in particular, are likely driven by the need to trace and plan paths through a maze; routines

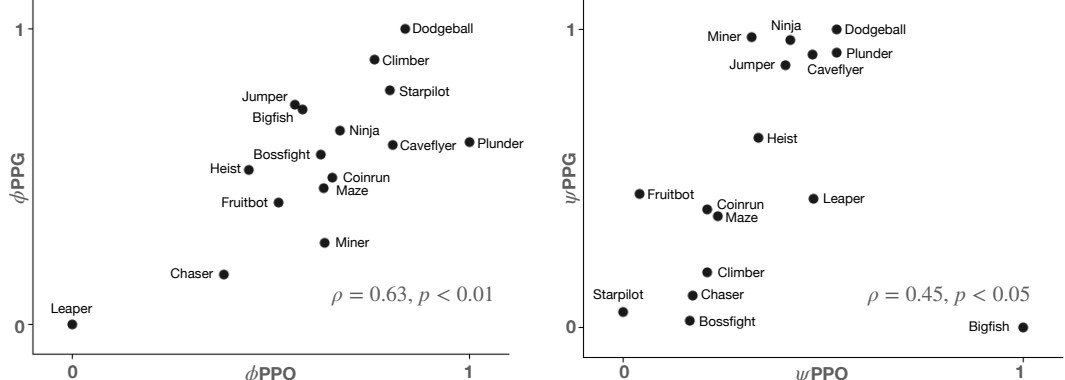

Figure 4: **LCD measurements of the perceptual and reinforcement learning challenges in *Procgen* translate across dRL algorithms.** Measurements of visual challenges $\phi$ (left) and reinforcement learning challenges $\psi$ (right) faced by PPO agents were significantly correlated with the challenges of PPG agents, indicating that results derived from applying the LCD to simpler and lower-overhead algorithms can guide the development of agents pushing the state-of-the-art.

which are known to be challenging for neural networks and draw upon feedback mechanisms in humans (Linsley et al., 2018a; Roelfsema et al., 2000; Roelfsema, 2006a; Ullman, 1984).

## 4   VALIDATING LCD TAXONOMIES BY BUILDING BETTER AGENTS

Our general approach for validating the computational taxonomy of the LCD is to demonstrate that the gains of targeted solutions to the challenges coincide with the LCD's predictions. We develop separate solutions for perceptual challenges and reinforcement learning challenges and describe those methods and results here.

**Developing solutions to perceptual challenges.**   We began by constructing a new perceptual front-end that we believed could address the perceptual challenges of *Procgen* games. Inspired by the reliance of humans and other animals on optic flow to effectively and efficiently navigate through their environments (Warren et al., 2001; Warren, 2021), we turned to motion perception as the base inductive bias for our agents. Specifically, we rely on the motion of objects in the world to extract perceptual groups that could instruct appropriate behavior.

It has been found that training DNNs to predict the optic flow between successive frames of video can induce the ability to segment object-like superpixels from complex scenes (Liu et al., 2021). Here, we build off this prior work to develop a self-supervised pre-training strategy for solving perceptual challenges in *Procgen*. Our architecture consists of three separate DNNs: (*i*) a ResNet18 (He et al., 2015) feature pyramid network (FPN) (Lin et al., 2016) for predicting the optic flow (*Flow encoder*) between two frames from successive timepoints of a sequence, frame $t$ and frame $t + 1$, (*ii*) a biologically inspired recurrent neural network (Linsley et al., 2018a) (*Object encoder*) for encoding the contents of frame $t$, and (*iii*) a fully convolutional (Long et al., 2015) decoder that learns how to use the two encoders' outputs to match frame $t$ to frame $t + 1$ via a differentiable warping module.

This model was pre-trained to learn visual representations in a purely self-supervised manner, by minimizing the mean squared error between two successive frames sampled from the seven games of *Procgen* which had consistent motion (Chaser, Leaper, Dodgeball, Climber, Bossfight, Starpilot, and Bigfish). We sampled frames for training from 14,000 videos (2,000 per game) in batches of 4 for $50k$ iterations using the Adam optimizer (Kingma & Ba, 2014) and a learning rate of $1e - 4$. As an alternative to our approach for self-supervision, we also evaluated the effectiveness of state-of-the-art features for one-shot recognition from a clip-ViT-B-32 model pre-trained on 400M natural images and captions (Radford et al., 2021).

**Perceptual challenge evaluations.**    We evaluated the LCD predictions of the perceptual challenges $\phi$ in each game by testing if preprocessing frames with our self-supervised front-end improved agent performance more for games with higher values of $\phi$ (*i.e.*, the front-end was fixed and not trained with reward). Because our self-supervised front-end relied on the game motion to extract perceptual groups, we focused our analysis on the same seven games with dynamic elements that it was pre-trained on. As predicted by the LCD, our front-end improved performance significantly more for perceptually challenging games (hard $\phi$) than perceptually simple games (easy $\phi$; Fig. 3b). Overall, the benefit of this front-end correlated significantly with the $\phi$ predicted for each game (Spearman's $\rho = 0.79$ between $\phi$ and the AUC of performance curves, $p < .05$). To understand the extent that our results merely demonstrate the value of pre-trained visual representations, we repeated our analysis while using a pre-trained CLIP as the perceptual front-end for agents. But while CLIP embeddings yield state-of-the-art performance in one-shot classification (Radford et al., 2021), they did not help agents learn more effective policies, faster (Appendix Fig. 12). In other words, LCD's perceptual challenges are more readily addressed by algorithmic solutions that induce object-like representations.

**Guiding algorithmic development with the LCD.** Given the success of our self-supervised perceptual front-end, an obvious approach towards building better agents is to simply use this front-end on every game. However, the impact of our front-end on performance was significantly blunted when measured on all games (the difference between green bars in Hard $\phi$ versus All Games, Fig. 3b). This finding points to a larger problem in algorithmic development for RL: without a reliable approach for diagnosing the specific computational challenges in a game, the effectiveness of reasonable solutions can be blunted or even appear to hurt agents, as our front end nominally does here.

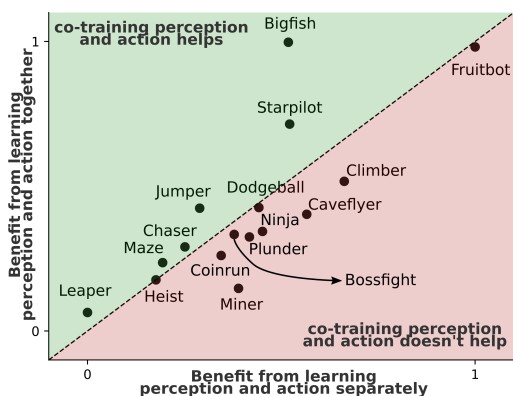

Figure 5: **A subset of games in *Procgen* see a synergistic benefit from co-training perception and action/policies end-to-end.** We compared the magnitude of improvement over baseline (PPO/pixels) agents experienced when they were given an ideal reinforcement learning algorithm (PPG) and perceptual representations (semantic segmentation) to the additive improvement of an agent given PPG over pixels and an agent given PPO and semantic segmentation inputs. Only 6/16 games benefit from co-training.

**Reward shaping for resolving reinforcement learning challenges.** The reinforcement learning challenges revealed by the LCD are due to the sparsity of rewards available to agents. As a partial solution to this problem, we investigated the impact of "reward shaping" (Skinner, 1938) on performance. Reward shaping is a technique from animal training where supplemental rewards are provided to make the learning problem easier Wiewiora (2010). We adopted this for three games amenable to reward shaping: Heist, Leaper, and Maze; the latter two LCD identified as having significant reinforcement learning challenges.

In Heist, agents must collect three keys and unlock their matching doors before they can reach the rewarding gem. We provided agents with additional rewards when they collected a key or opened a door. In Leaper, agents must safely get across several lanes of roads followed by several rows of rivers. Lanes have moving cars agents must avoid, while rivers have moving logs agents must stay on. We gave agents additional rewards whenever they reached a new lane or river.

In Maze, agents must wander through a complex maze to get to a reward of cheese; however, the topological complexity of the mazes can deem rewards difficult to discover. To address this problem, we encourage exploration of novel locations through intermediate rewards Bellemare et al. (2016); Ecoffet et al. (2021). In all of these cases, agents did not get these auxiliary rewards during test time.

**Reinforcement learning challenge evaluations.**    Agents trained with reward shaping on Heist ($t = 1.69, p < .05$), Leaper ($t = 41.03, p < .001$), and Maze ($t = 2.69, p < .01$) performed signifi-

cantly better than those trained on the normal versions of each game (all tests are one-tailed $t$-tests; Appendix Fig. 13). Moreover, the improvements in performance from reward shaping significantly correlated with the $\psi$ values of each game tested here (Spearman's $\rho = 0.92$, $p < .001$).

**The LCD-derived taxonomy translates across dRL algorithms.**   Our findings thus far indicate that an LCD calibrated with PPO agents makes reliable predictions about the challenges of individual games in *Procgen*. But despite the popularity of PPO, it has been surpassed by more recent dRL algorithms, like PPG, which achieved state-of-the-art performance on *Procgen*. To what extent does the LCD taxonomy derived from PPO translate to more effective dRL algorithms? We tested this question by applying the LCD to agents trained with PPG (Appendix Figs. 10, 11) and found that its predictions were significantly correlated with those from an LCD calibrated on PPO agents (Fig. 4). This means that our PPO-derived taxonomy can be relied on prospectively for algorithmic development on *Procgen*.

**The whole is only sometimes greater than the sum of the parts in dRL.**   Training perception-for-action, end-to-end, is the standard approach to dRL ever since the introduction of DQNs. Here, we tested whether agents trained on *Procgen* experienced a superlinear benefit of co-training perception and action or not. We did this by computing two performance differentials. In both cases, the *baseline* was a PPO agent trained on pixels. (*i*) the performance of a PPG agent trained on semantic segmentation inputs *minus* baseline, and (*ii*) the performance of a PPG agent trained on pixels *minus* baseline *plus* the performance of a PPO agent trained on semantic segmentation inputs *minus* baseline (Fig. 5).

Co-training perception for action helped performance in 6 of 16 games: Leaper, Maze, Chaser, Jumper, Starpilot, and Bigfish. Co-training did not help in 10 of 16 games: Heist, Dodgeball, Fruitbot, Miner, Coinrun, Bossfight, Plunder, Ninja, Caveflyer, and Climber. This finding poses an intriguing prospect. It suggests the existence of different "minibenchmarks" within *Procgen* that researchers should tap into, depending on whether they wish to quantify "improvements" to an agent's policy learning algorithm, its representation learning system, or synergies between the two.

## 5  DISCUSSION

Ever since the introduction of the DQN, the field of dRL has consistently succeeded in reaching and exceeding human performance on defined tasks: beating grand masters at the game of Go, or exceeding the average performance of humans on the Atari benchmark. However, even in these successes, it was clear that certain games presented different challenges than others. Take "Montezuma's Revenge", which the original DQN could not solve. Subsequent analysis of Montezuma's Revenge led to the conclusion that the failures of the DQN were because of its long time horizon and sparse rewards; challenges which, these days, have positioned it as a critical test for dRL algorithms (Ecoffet et al., 2019; Roderick et al., 2018; Salimans & Chen, 2018). We believe that the field of dRL can benefit from a systematic approach to diagnosing such computational challenges in any benchmark. Our LCD tool is a major step towards this goal.

By applying our LCD to the *Procgen* benchmark, we discover a novel taxonomy of those games, organized according to the relative perceptual and reinforcement learning challenges of each. This taxonomy delivers two prescriptions for algorithmic development in dRL. First, because the computational challenges of games in benchmarks like *Procgen* are not i.i.d., there will be low signal for adjudicating between algorithms designed to improve a problem that is not over-represented in the benchmark. Second, although humans are capable of learning perception-for-action, and dRL agents generally learn in this way, not all games in benchmarks like *Procgen* benefit from doing so. In the short-term, these problems can be addressed by using the LCD to select games in a benchmark that are aligned with the algorithmic design goals of researchers. In the long-term, there is a need in the field for new benchmarks that can tap into a wider range of computational challenges with less bias than is found in *Procgen*. We believe that our LCD is an important tool for advancing the pace of progress in dRL, and we release our experimental code to support this goal.

## 6 ETHICS STATEMENT

Our work is a contribution to deep reinforcement learning and so inherits the concerns already inherent in dRL applications. Currently, our work has been applied to harmless toy videogame tasks, but improper application of dRL to real world applications can have negative externalities on society, especially when algorithms make poor decisions. Depending on the domain of application, these can include discriminatory decisions against marginalized groups or serious injury and fatality if control of vehicles or critical systems are involved. Advancing agents' ability to make better decisions would help mitigate this issue and eventually broaden the domains in which dRL can help, and our contribution of identifying limitations in current dRL agents will help in that regard.

## 7 REPRODUCIBILITY STATEMENT

We have taken several steps to ensure the reproducibility of our work. We have made all code for our modified *Procgen* environment and the self-supervised model for perceptual grouping available at `https://anonymous.4open.science/r/lcd-procgen/`. For PPO, we have used the code available at `https://github.com/openai/train-procgen` along with the hyperparameters found in Cobbe et al. (2020). For PPG, we have used the code at `https://github.com/openai/phasic-policy-gradient` and the hyperparameters in Cobbe et al. (2021). Finally, we have reported the computing resources used and the number of experiments run under "dRL algorithms" of Section 3.

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

## A APPENDIX

### A.1 NETWORK ARCHITECTURE

Following Cobbe et al. (2020), we used the convolutional actor-critic IMPALA architecture described in Espeholt et al. (2018). This is a deeper network than what has been used previously, consisting of 15 convolutional layers, 16 residual blocks, and 1.6M parameters. We refer the reader to (Espeholt et al., 2018) for details. For raw pixel, semantic segmentation, and figure-ground segmentation, the number of input channels was fixed to three. This controls the model architecture to be identical regardless of input. For semantic or figure-ground masks, we replicated the masked input three times and fed that into each of the three convolutional input channels. Our self-supervised visual model outputs 16 channel perceptual masks (at the same spatial resolution of the inputs) which are concatenated with the RGB values and fed into the actor-critic network for policy learning.

### A.2 REWARD NORMALIZATION

Given that each task involves varying reward magnitudes, episode rewards $r$ were normalized to facilitate comparison across tasks. Normalized rewards $r'$ were computed as $\frac{r - R_{min}}{R_{max} - R_{min}}$, where $R_{max}$ is the theoretical maximum score attainable in an episode while $R_{min}$ is the average score a random agent would attain.

## A.3 STOCHASTIC REWARD FEEDBACK

Here, we prove that the expected return under policy $\pi$ does not change under our manipulation of reward probability. We proceed in two parts. First, we show that the expected reward received from a rewarding event remains unchanged. Then we show that the action-value function $Q^\pi(s, a)$ for policy $\pi$ remains unchanged. By implication, this means the optimal policy does not change either.

Let $R(s, a)$ denote the original reward function at state $s$ and action $a$. In our manipulation, rewards are generated stochastically with probability $p(s, a)$, but reward magnitudes are rescaled by $\frac{1}{p(s,a)}$.

**Definition A.1** (Modified reward function). The modified reward function $\tilde{R}(s, a)$ is given by

$$\tilde{R}(s, a) \triangleq \frac{R(s, a)}{p(s, a)} \eta(s, a),$$

where $\eta(s, a) \sim Bern[p(s, a)]$.

**Lemma A.1.** *The expected value of $\tilde{R}(s, a)$ is $R(s, a)$.*

*Proof.*

$$
\begin{aligned}
\mathbb{E}_\eta \left[ \tilde{R}(s, a) \right] &\triangleq \mathbb{E}_\eta \left[ \frac{R(s, a)}{p(s, a)} \eta(s, a) \right] \\
&= \frac{R(s, a)}{p(s, a)} \mathbb{E}_\eta \left[ \eta(s, a) \right] \\
&= R(s, a),
\end{aligned}
$$

since $\mathbb{E}_\eta \left[ \eta(s, a) \right] = p(s, a)$. $\qquad\square$

Using this lemma, we now prove our main result. Let $Q^\pi(s, a)$ denote the action-value function induced by the original reward function $R(s, a)$ while following policy $\pi$, and let $\tilde{Q}^\pi(s, a)$ denote the action-value function induced by $\tilde{R}(s, a)$ while following the same policy.

**Theorem A.2.** *The action-value function $\tilde{Q}^\pi(s, a)$ is given by $Q^\pi(s, a)$.*

*Proof.* As is common practice in RL, we shall discount future rewards by $\gamma \in [0, 1]$ on each future time step. Before beginning, we shall introduce some notation. We shall denote the current time step by $t$. We shall denote the state and action at any time $\tau$ by $s_\tau$ and $a_\tau$. Finally, by an abuse of notation, we shall use $\mathbb{E}_\pi[\cdot]$ to denote expectations over state-action trajectories generated by both the policy $\pi$ and the state-transition probability $p(s'|s, a)$, where $s'$ is the successor state.

We now proceed to the proof.

$$
\begin{aligned}
\tilde{Q}^\pi(s, a) &\triangleq \mathbb{E}_{\pi,\eta} \left[ \sum_{k=0}^\infty \gamma^k \tilde{R}(s_{t+k+1}, a_{t+k+1}) \;\middle|\; s_t = s, a_t = a \right] \\
&= \mathbb{E}_\pi \left[ \sum_{k=0}^\infty \gamma^k \mathbb{E}_\eta \left[ \tilde{R}(\tilde{s}, \tilde{a}) \;\middle|\; \tilde{s} = s_{t+k+1}, \tilde{a} = a_{t+k+1} \right] \;\middle|\; s_t = s, a_t = a \right] \quad \text{(law of total expectation)} \\
&= \mathbb{E}_\pi \left[ \sum_{k=0}^\infty \gamma^k R(s_{t+k+1}, a_{t+k+1}) \;\middle|\; s_t = s, a_t = a \right] \quad \text{(Lemma A.1)} \\
&= Q^\pi(s, a)
\end{aligned}
$$

$\square$

Note that in our experimental manipulations, we used a single reward probability $p$ that was independent of $s$ and $a$.

## A.4 Performance change with increasing credit assignment difficulty

To quantify the average change with increasing credit assignment difficulty, we first computed the reward curve AUC $AUC(p)$ for $p = 0.25, 0.5, 0.75, 1.0$. We next computed the absolute change in AUC between successive values of $p$; that is, $|AUC(p = 1.0) - AUC(p = 0.75)|$, $|AUC(p = 0.75) - AUC(p = 0.5)|$, and $|AUC(p = 0.5) - AUC(p = 0.25)|$. Finally, we averaged over these measures to get our final measure of average change.

## A.5 Supplementary Figures

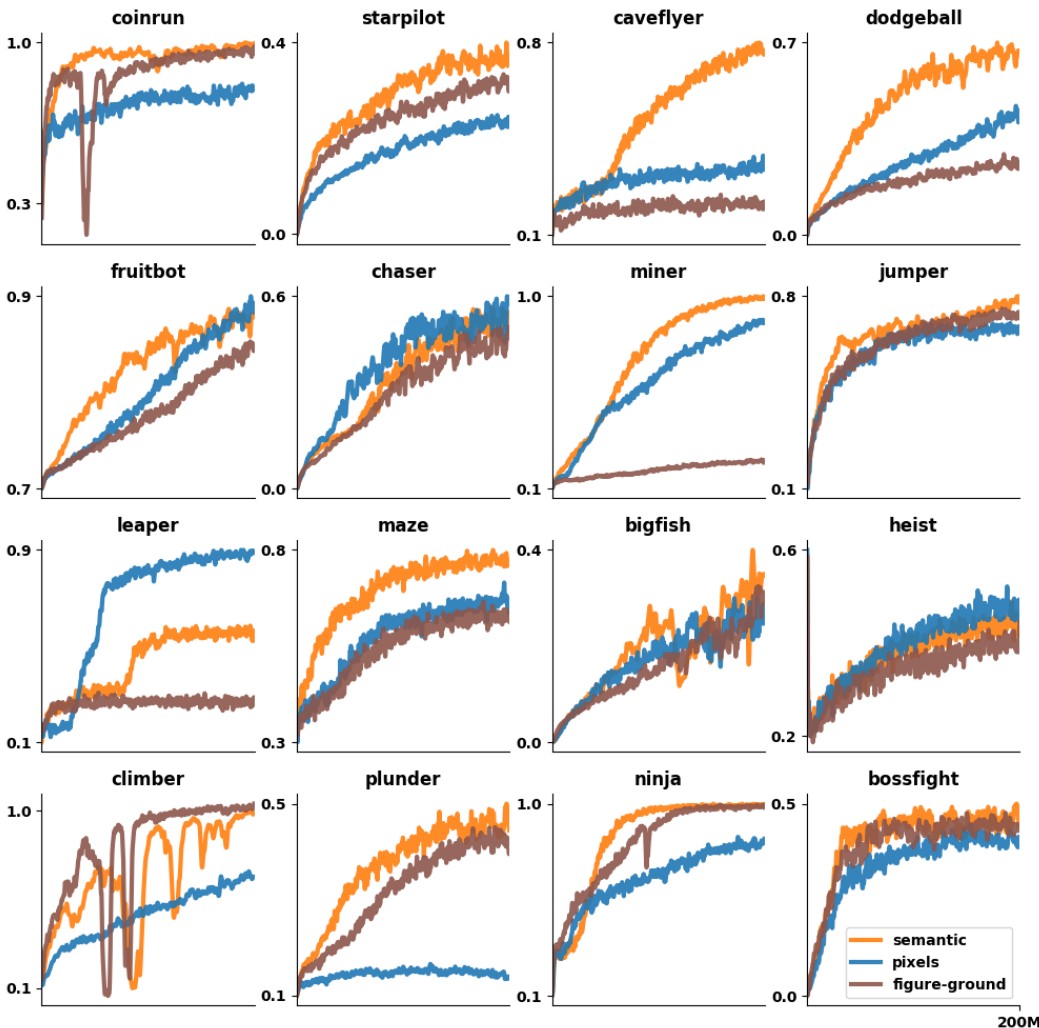

Figure 6: **Normalized reward curves while systematically varying perceptual complexity.** X-axis denotes the number of "interactions" an agent performs with its environment during training and Y-axis denotes normalized rewards. We trained all these agents for a total of 200M steps.

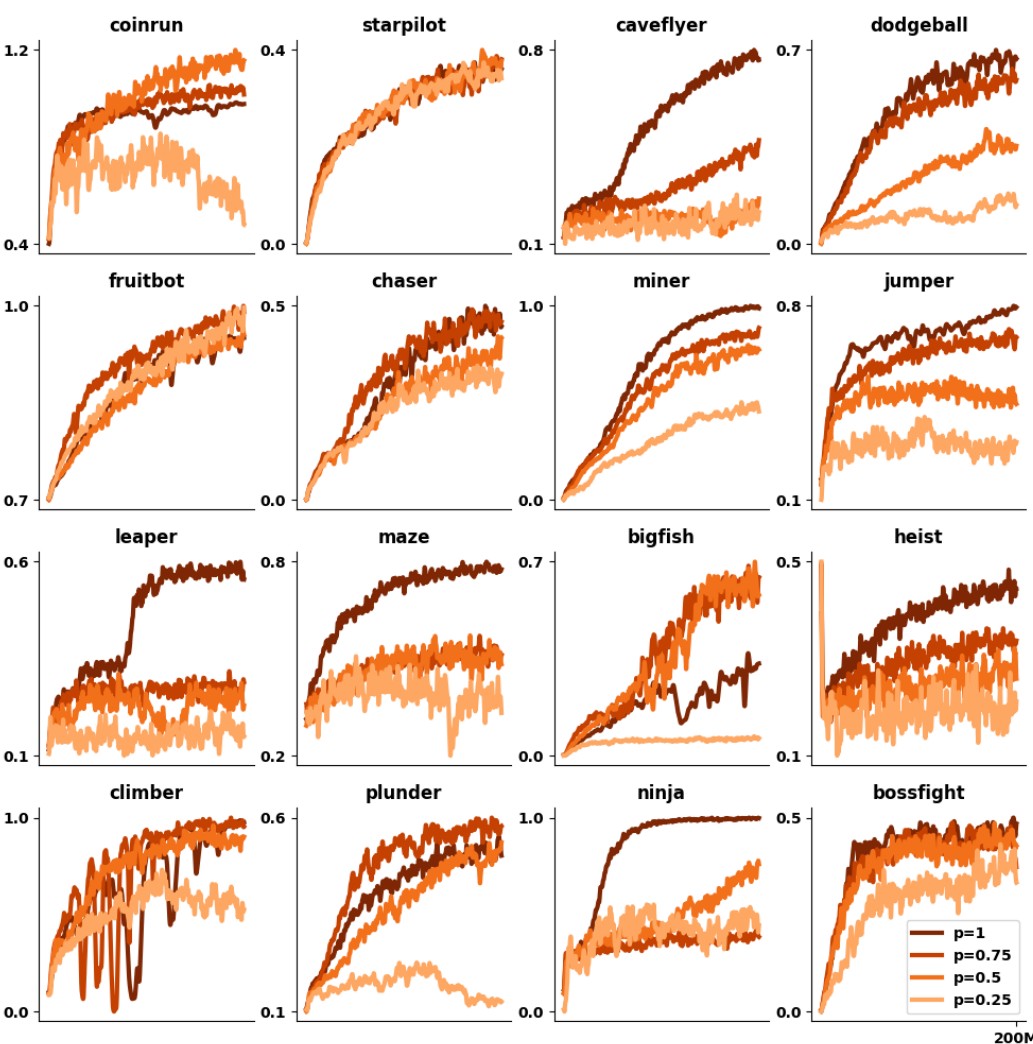

Figure 7: **Normalized reward curves while systematically varying reward stochasticity.** X-axis denotes the number of "interactions" an agent performs with its environment during training and Y-axis denotes normalized rewards. We trained all these agents for a total of 200M steps. The perceptual input to these agents were semantic representations. While performing evaluations, the reward mechanisms were set to the original configuration thus effectively only testing the impact of these perturbations on policy learning.

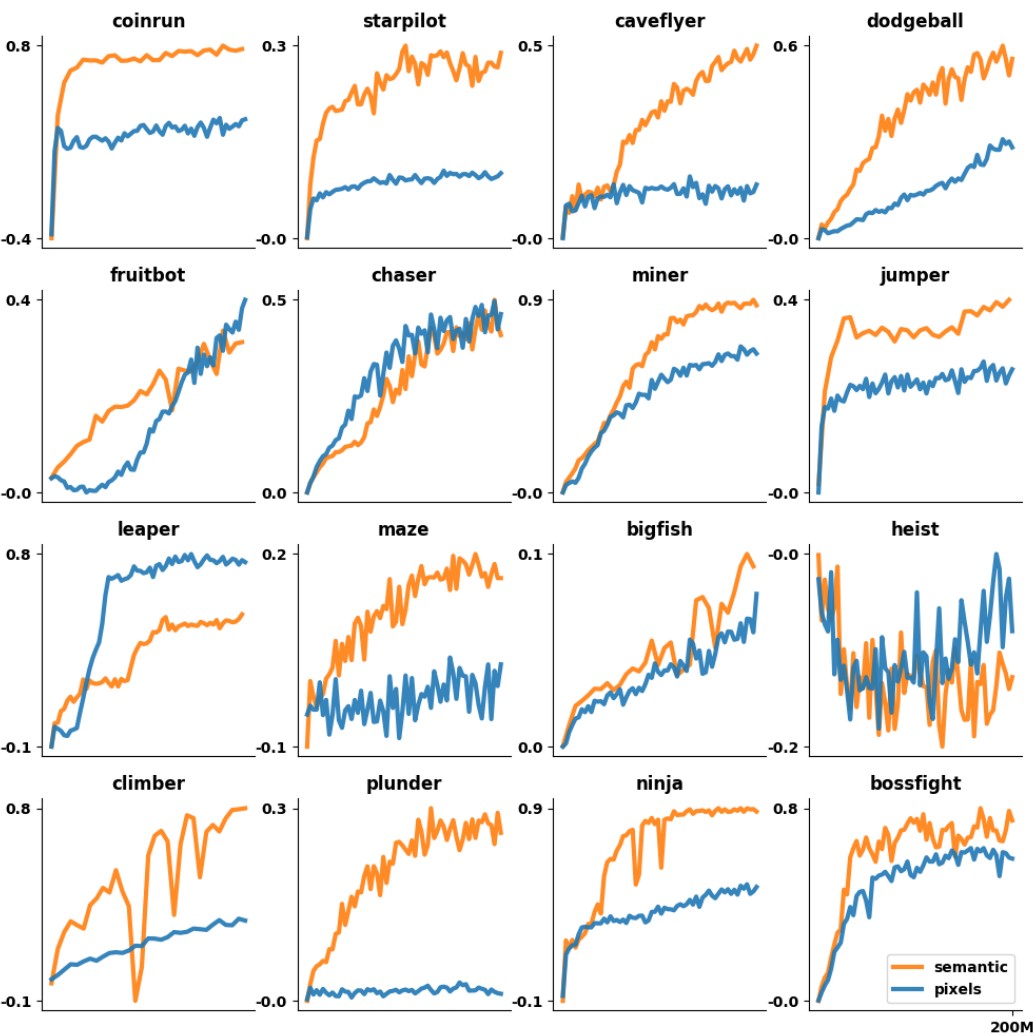

Figure 8: **Normalized reward curves, on *i.i.d. generalization* levels, while systematically varying perceptual complexity.** X-axis denotes the number of "interactions" an agent performs with its environment during training and Y-axis denotes normalized rewards. We trained all these agents for a total of 200M steps. Agents were evaluated approximately every $60K$ steps.

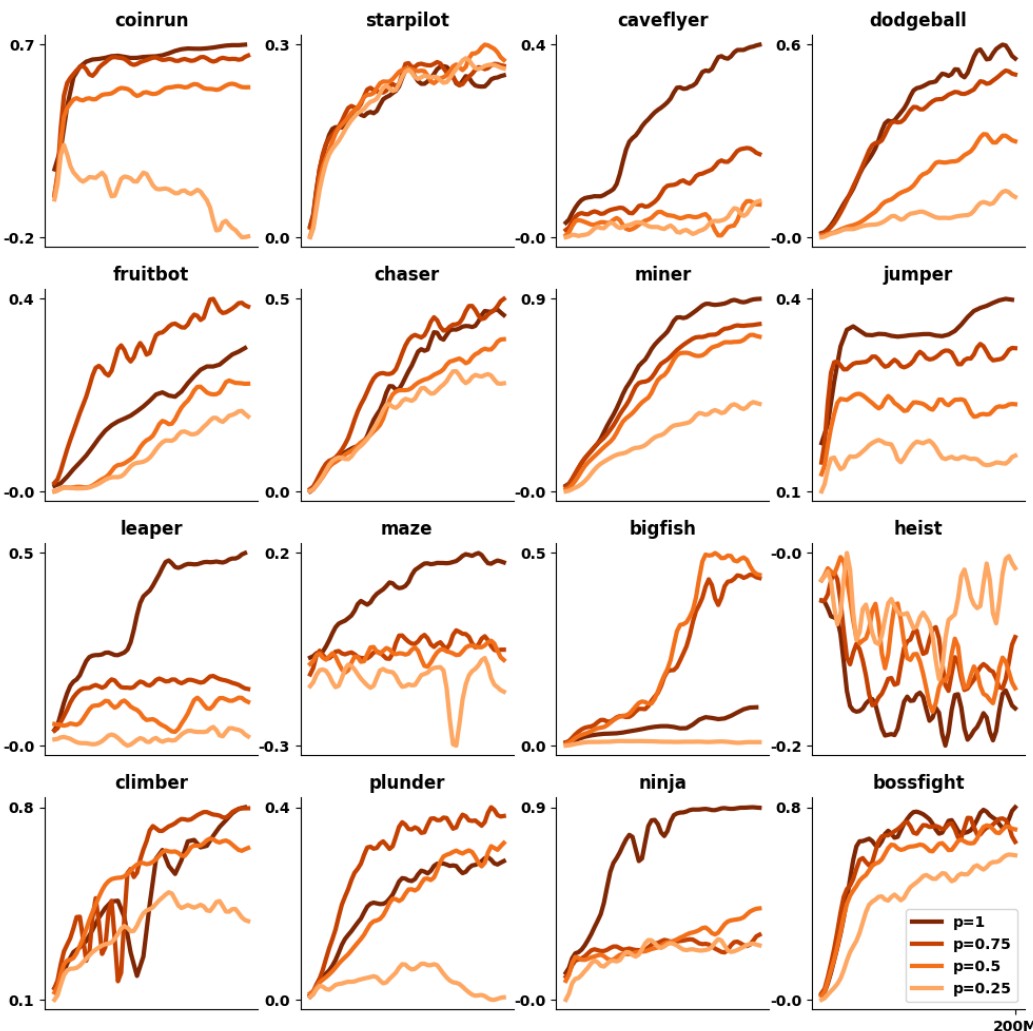

Figure 9: **Normalized reward curves, on *i.i.d. generalization* levels, while systematically varying reward stochasticity.** X-axis denotes the number of "interactions" an agent performs with its environment during training and Y-axis denotes normalized rewards. We trained all these agents for a total of 200M steps. The perceptual input to these agents were semantic representations. Agents were evaluated approximately every $60K$ steps. While performing evaluations, the reward mechanisms were set to the original configuration thus effectively only testing the impact of these perturbations on policy learning.

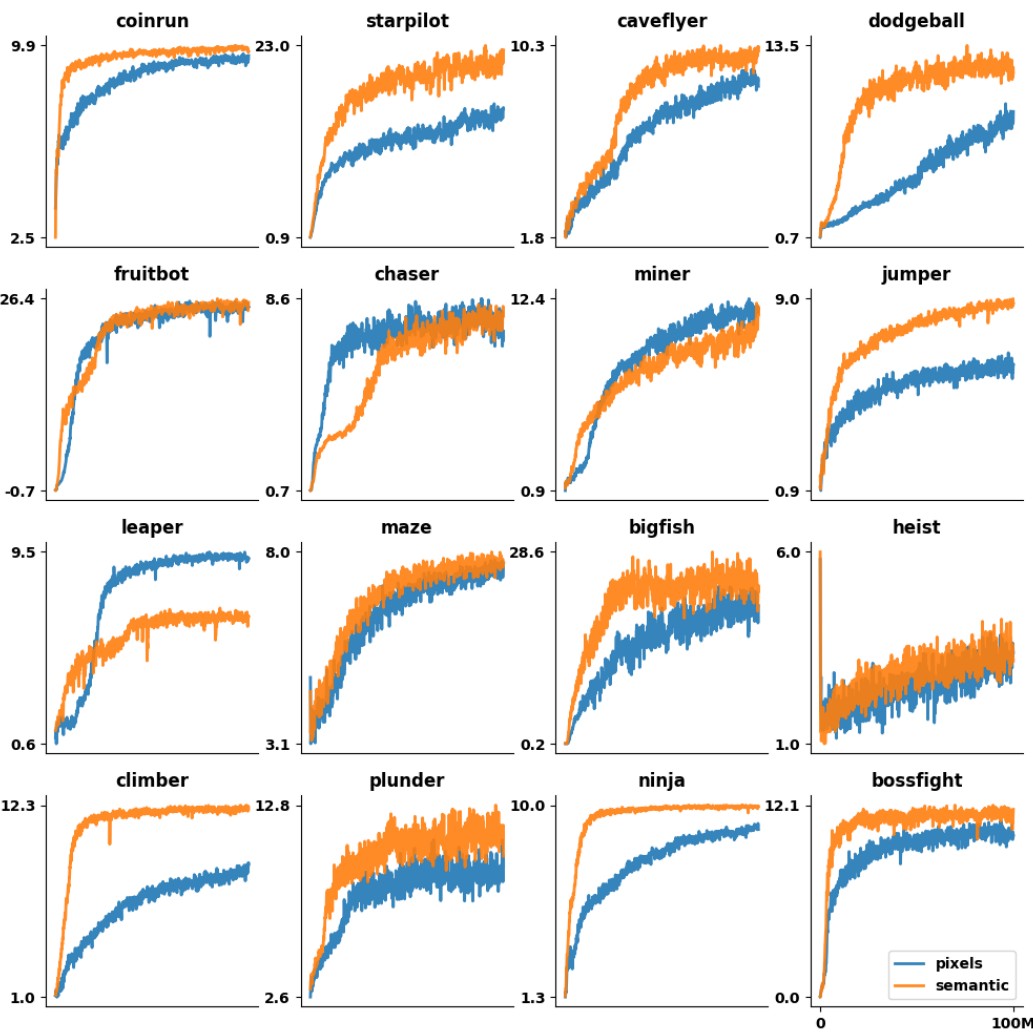

Figure 10: **Training dRL agents with varying levels of perceptual complexity using the proximal policy gradient algorithm.** X-axis denotes the number of "interactions" an agent performs with its environment during training and Y-axis denotes rewards on training levels. We trained these agents for a total of 100M steps.

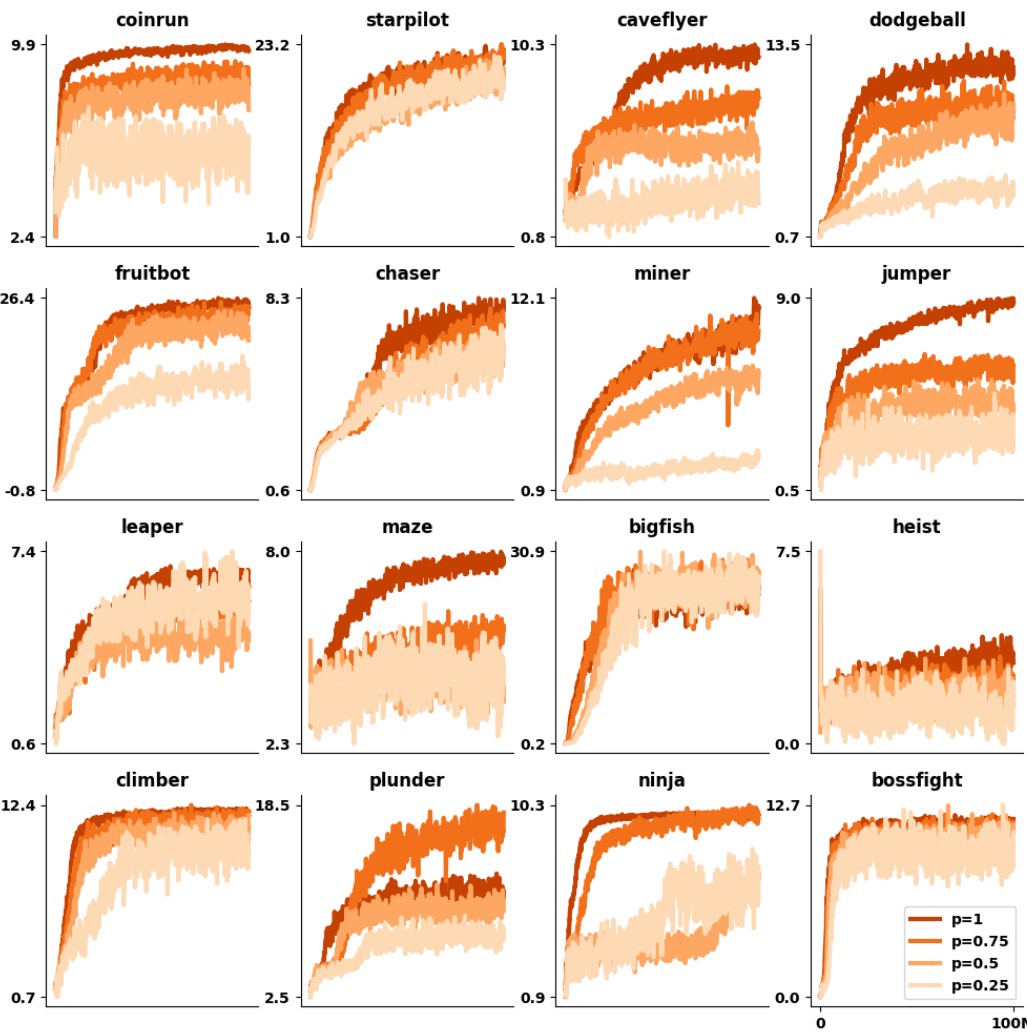

Figure 11: **Training dRL agents with varying levels of stochastic feedback using the proximal policy gradient algorithm.** X-axis denotes the number of "interactions" an agent performs with its environment during training and Y-axis denotes rewards on training levels. We trained these agents for a total of 100M steps. The perceptual inputs to these agents were semantic representations. While performing evaluations, the reward mechanisms were set to the original configuration thus effectively only testing the impact of these perturbations on policy learning.

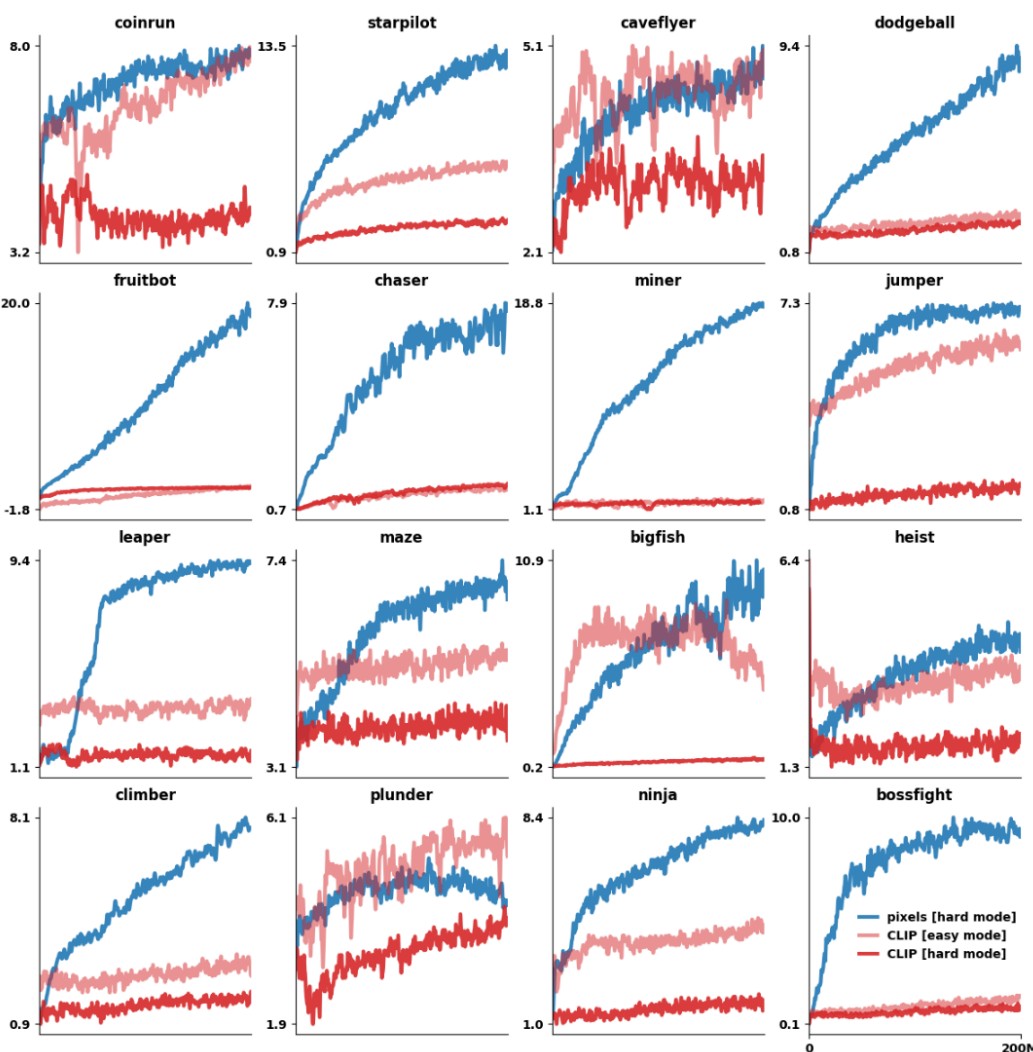

Figure 12: **Policy learning experiments in dRL agents operating on visual representations learned from natural language supervision.** We test the extent to which generalist visual representations support policy learning. We find that while CLIP Radford et al. (2021), a transformer-based architecture, is state-of-the-art on zero-shot image classification, its representations do not support policy learning adeptly. We train agents on both the "easy" and "hard" versions of *Procgen*. While our agent was able to learn certain tasks in the "easy" mode, they predominantly struggled in the "hard" mode. X-axis denotes the number of "interactions" an agent performs with its environment during training and Y-axis denotes rewards. We trained these agents for a total of 200M steps.

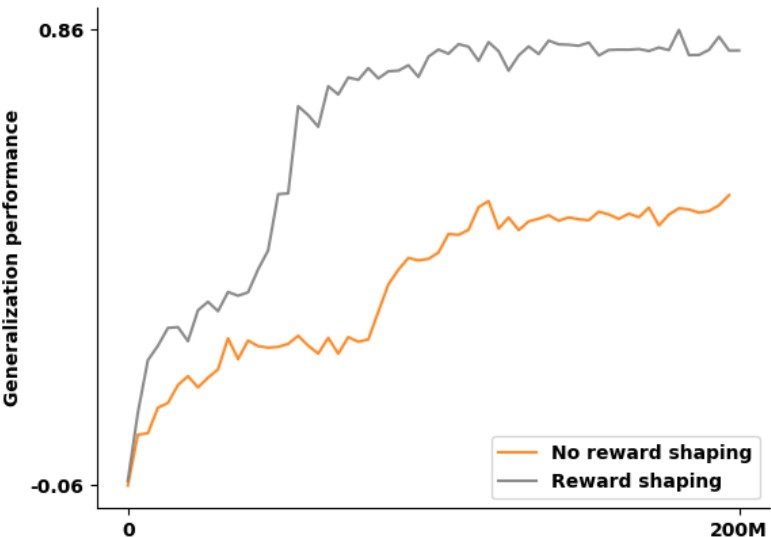

Figure 13: **The effect of "reward shaping" on Leaper – a high $\psi$ task as predicted by our taxonomy.** In the traditionally challenging reinforcement learning problem we find that encouraging agents to discover sub-goals significantly improved both the sample efficiency and generalization capacity ($t = 41.03, p < .001$). X-axis denotes the number of "interactions" an agent performs with its environment during training and Y-axis denotes normalized rewards. We trained both these agents on semantic representations for a total of 200M steps. While performing evaluations, we revert to the original reward scheme to remain compatible for comparing to the "naive" model.

A.6 SPECIFICS OF OUR PERCEPTUAL PARAMETERIZATIONS OF THE *Procgen* BENCHMARK

| Image | Game | Mask Details |
|---|---|---|
|  | `bigfish` | There are two entity IDs: (1) the player, and (2) all other fish. |
|  | `bossfight` | There are seven entity IDs: (1) the player, (2) player projectiles, (3) the enemy, (4) enemy projectiles, (5) the enemy's shield, (6) meteors, and (7) explosions. |
|  | `caveflyer` | There are eight entity IDs: (1) the player, (2) enemy ships, (3) player projectiles, (4) targets, (5) the goal landing pad, (6) meteors, (7) explosions, and (8) the surrounding terrain. |
|  | `chaser` | There are seven entity IDs: (1) the player, (2) enemies, (3) enemies in their weakened state, (4) enemy eggs, (5) small collectible squares, (6) large collectible stars, and (7) the walls. |
|  | `climber` | There are four entity IDs: (1) the player, (2) enemies, (3) the star rewards, and (4) the surrounding walls, ground, and platforms. |
|  | `coinrun` | There are eight entity IDs: (1) the player, (2) enemies, (3) enemy barriers, (4) saw obstacles, (5) lava obstacles, (6) the goal star, (7) crates, and (8) the surrounding walls and ground. |
|  | `dodgeball` | There are seven entity IDs: (1) the player, (2) player projectiles, (3) the enemies, (4) enemy projectiles, (5) the locked exit, (6) the unlocked exit, and (7) the walls. |
|  | `fruitbot` | There are eight entity IDs: (1) the player, (2) player projectiles, (3) rewarding objects, (4) penalizing objects, (5) locks, (6) locked doors, (7) the goal, and (8) barriers and the terrain below and above the beginning and end of each episode, respectively. |

Table 1: **Outlining the "Semantic" organization of each environment in the *Procgen* benchmark.**

| Image | Game | Mask Implementation |
|---|---|---|
|  | heist | There are six entity IDs: (1) the player, (2-4) up to three unique keys and their corresponding locks, (5) the goal crystal, and (6) the walls. |
|  | jumper | There are six entity IDs: (1) the player, (2) spikes, (3) the goal, (4) the needle of the compass and the bar indicating the distance to the goal, (5) the circle of the compass, and (6) the surrounding terrain. |
|  | leaper | There are six entity IDs: (1) the player, (2) cars, (3) the road, (4) log platforms, (5) the water, and (6) the finish line. |
|  | maze | There are three entity IDs: (1) the player, (2) the goal cheese, and (3) the walls. |
|  | miner | There are five entity IDs: (1) the player, (2) boulders, (3) crystals, (4) the exit, and (5) dirt. |
|  | ninja | There are seven entity IDs: (1) the player, (2) player projectiles, (3) the goal mushroom, (4) bombs, (5) explosions, (6) the bar indicating jump charge, and (7) the surrounding walls, ground, and platforms. |
|  | plunder | There are eight entity IDs: (1) the player and ally ships, (2) player projectiles, (3) enemy ships, including the cue ship, (4) the circle behind the cue ship, (5) barriers, (6) explosions, (7) the time-countdown status bar, and (8) the points-accrued status bar. |
|  | starpilot | There are eleven entity IDs: (1) the player, (2) player projectiles, (3) slow enemies flyers, (4) fast enemies flyers, (5) enemy flyer projectiles, (6) enemy turrets, (7) enemy turret projectiles, (8) meteors, (9) clouds, (10) the finish line, and (11) explosions. |

Table 2: **Outlining the "Semantic" organization of each environment in the *Procgen* benchmark.**

| Image | Game | Mask Details |
|---|---|---|
|  | bigfish | All entities share the same ID. |
|  | bossfight | All entities share the same ID. |
|  | caveflyer | All entities share the same ID. |
|  | chaser | All entities share the same ID. |
|  | climber | All entities share the same ID. |
|  | coinrun | All entities share the same ID. |
|  | dodgeball | All entities share the same ID. |
|  | fruitbot | All entities share the same ID. |

Table 3: **Outlining the "Figure/Ground" organization of each environment in the *Procgen* benchmark.**

| Image | Game | Mask Implementation |
|---|---|---|
|  | heist | There are four entity IDs: (1-3) up to three unique keys and their corresponding locks, and (4) all other entities. |
|  | jumper | There are three entity IDs: (1) the needle of the compass and the bar indicating the distance to the goal, (2) the circle of the compass, and (3) all other entities. |
|  | leaper | There are two entity IDs: (1) the road and water, and (2) all other entities. |
|  | maze | All entities share the same ID. |
|  | miner | All entities share the same ID. |
|  | ninja | There are two entity IDs: (1) the bar indicating jump charge, and (2) all other entities. |
|  | plunder | There are five entity IDs: (1) the player and friendly ships, (2) enemy ships, including the cue ship, (3) the time-countdown status bar, (4) the points-accrued status bar, and (5) all other entities. |
|  | starpilot | All entities share the same ID. |

Table 4: **Outlining the "Figure/Ground" organization of each environment in the *Procgen* benchmark.**

## A.7 IMPROVEMENTS IN GENERALIZATION PERFORMANCE ACROSS EASY AND HARD $\phi$ TASKS

| Easy ($\phi$) % improvement | Chaser 7.5% | Leaper $-117.6\%$ | Bigfish 37.9% | Bossfight $-39.8\%$ |
|---|---|---|---|---|
| Hard ($\phi$) % improvement | Starpilot 28.7% | Dodgeball 71.4% | Climber 3364.6% | – |

