# OpenReview forum: "Diagnosing and exploiting the computational demands of videos games for deep reinforcement learning"
_ICLR.cc/2023/Conference — Submitted to ICLR 2023_

### Official Review · Reviewer_Ni8m · 2022-10-20

**Confidence:** 4
**Correctness:** 4
**Technical Novelty And Significance:** 3
**Empirical Novelty And Significance:** 2
**Recommendation:** 5

**Clarity, Quality, Novelty And Reproducibility:**

**[Clarity]**

The paper is basically clear and well-written. The plots are vivid.

**[Quality]**

As an empirical study, the quality is mostly OK, while there is some space to improve (see above).

**[Novelty]**

The proposed approach is novel to my knowledge and it touches a vital problem in deep RL.

**[Reproducibility]**

The source code is available, while I have not checked it.

**Strength And Weaknesses:**

**[Strength]**
- The paper tackles an important yet unclear question in deep RL (disentangling the challenges from representation learning and reward-maximization).
- The proposed LCD is a practical and effective way to numerically address the question in Procgen environments.
- The authors give useful suggestion to improve deep RL performance when LCD is known.

**[Limitations]**
- The way of obtaining LCD is hindsight, it is still unclear where the challenge locates before conducting RL.
- The way of computing LCD is not general, which requires segmentation of image input already known.
- The computational cost of LCD is high (require repeating the whole training process for several times).

**[Suggestions]**
- The way of measuring the reinforcement learning challenge could be more compresensive: the difficulty in RL does not solely come from sparse reward (e.g., in MuJoCo tasks, the reward is non sparse, but it is still sometimes challenging).
- I would like to see more experimental results besides the ProcGen benchmark.




**Summary Of The Paper:**

The paper tackled with disentangling the challenges from representaion learning and policy learning in deep RL for vision-based input. The main contribution is the introduction of "Learning Challenge Diagnosticator" (LCD), which is metric to measure the aforementioned problem. LCD is computed by examing the performance change when perturbating the vision input and the reward signal, respectively. The authors did experiments in 16 Procegen game environments to support their claims. Furthermore, for different challenges in deep RL, the paper suggested to use pre-trained representation learning and reward-shaping to overcome the two challenges respectively.

**Summary Of The Review:**

This study aims to address an increasingly significant question in deep RL, that is disentangling the difficulty of representation learning and policy learning in deep RL with image observations. The paper put a step forward in this direction, while the proposed method has some room to improve. I encourage the authors to update their method by overcoming the limitations I mentioned above, in which case the work will be a fantastic contribution to the deep RL society. However, currently I vote for a weak rejection.

---

> ### Author Response · Authors · 2022-11-19
> **Response**
>
> The way of obtaining LCD is hindsight, it is still unclear where the challenge locates before conducting RL. The way of computing LCD is not general, which requires segmentation of image input already known. The computational cost of LCD is high (require repeating the whole training process for several times).
>
> The LCD is in fact computationally cheap to compute: as we show in **Fig. 4**, the LCD taxonomy derived from an agent trained with PPO generalizes to an agent trained with more performant algorithms, like PPG (which is state-of-the-art on Procgen). The LCD need not be recomputed for Procgen and the computational cost is now amortized. The taxonomy and validative results we provide can guide algorithmic development on Procgen.
>
> It is true that the LCD requires segmentation masks to be computed. This means that the method generalizes to any video game that can be modified. We leave extending it to real-world tasks to future work.
>
> > The way of measuring the reinforcement learning challenge could be more comprehensive: the difficulty in RL does not solely come from sparse reward...
>
> Our LCD acts as a sensitivity analysis, where the perceptual or RL problem in a game is perturbed while making the other problem as structured and easy-to-learn as possible. As we wrote, the important decision is to “...begin by modifying a game to put its perceptual representations and reinforcement learning problem under experimental control.“ For the RL problem, we considered reward sparsity because it has been found to degrade learning signal, making it difficult for RL agents to learn anything [1, 2, 3]. Arbitrarily changing the reward function is fraught with the risk of changing the task’s optimal policy, and we prove in *SI A.3* that our manipulation decreases the learning signal’s saliency without altering the optimal policy.
>
> We have expanded our discussion to expand on the philosophy behind perceptual/RL perturbations, and what motivated our choices for the experiments in this manuscript.
>
> > I would like to see more experimental results besides the ProcGen benchmark.
>
> We are in the process of extending the LCD to a novel RL challenge, Avalon, and will add those results to the manuscript as soon as they are finished [4].
>
> [1] Jaderberg M et al., 2017. Reinforcement learning with unsupervised auxiliary tasks. ICLR.
>
> [2] Houthooft R et al., 2016. VIME: Variational information maximizing exploration. NeurIPS.
>
> [3] Andrychowicz M et al., 2016. Hindsight experience replay. NeurIPS.
>
> [4] Albrecht J et al., 2022. Avalon: A benchmark for RL generalization using procedurally generated worlds. NeurIPS.

---

> ### Comment · Reviewer_Ni8m · 2022-11-23
> **Thanks for the reply**
>
> Thanks for the reply. The authors replied "We are also working on extending the LCD to another RL challenge, the recently released Avalon. We will update the manuscript with these experiments as soon as they are completed."
>
> However, I did not see the update. Therefore I make no change to my recommendation.

---

### Official Review · Reviewer_SeAm · 2022-10-20

**Confidence:** 4
**Correctness:** 3
**Technical Novelty And Significance:** 2
**Empirical Novelty And Significance:** 2
**Recommendation:** 3

**Clarity, Quality, Novelty And Reproducibility:**

* Clarity: Good.
* Quality and Novelty: Average. The results are unsurprising and the benefits are not clear.
* Reproducibility: Good

**Strength And Weaknesses:**

Strength:
* The paper is overall clear and easy to follow.
* The experiments on procgen are comprehensive and the results are statistically significant.

Comments:
* In Section 4, the authors choose 7 games for pretraining visual representations with self-supervision. I am wondering why these 7 games are chosen. In other words, I don't quite get what "consistent motion" means.
* The results in Figure 3 seem to be dominated by two games (Leaper and Climber). For other games, the improvements look similar across easy and hard $\phi$. Thus, I do not think the conclusion drawn from the results is convincing enough.
* In Page 8 (Reinforcement learning challenge evaluations), the authors state that "the improvements in performance from reward shaping significantly correlated with the values of each game tested here". However, it depends on the reward shaping methods used (different envs use different reward shaping). So it looks to me that such correlation does not imply anything meaningful.
* In "The whole is only sometimes greater than the sum of the parts in dRL" on Page 8, I do not fully understand the connection between the experiments setup (i) /(ii) and the implication of learning perception and action together/separately. Can the authors elaborate on it?
* In Figure 9, for some games, using $p < 1$ yields much better performance. Do the authors have any interpretation of such results?

Weaknesses:
* My major concern is about the insights offered by this paper. It is not surprising that some games require more perceptual learning while others require less. Even if we know (from LCD) that a particular game falls into the former category, can we totally disregards the RL part for this game? I guess the answer is probably no. So eventually, we still need to consider both perceptual learning and RL. In short, I think the benefits brought by LCD are not clear.

Minor Issues:
* The citation of Procgen benchmark is incorrect. It should be "Leveraging Procedural Generation to Benchmark Reinforcement Learning".
* For Figure 4 and Figure 5, it is better to keep the equal axis aspect ratio, i.e., making it a square instead of a rectangle.

**Summary Of The Paper:**

This paper introduces the Learning Challenge Diagnosticator (LCD), a tool for assessing the difficulties of perceptual learning and reinforcement learning for a video game environment. The authors apply LCD on the Procgen benchmark, and reveals that how much perceptual learning and reinforcement learning affects the performance for each game. Instead of having a single “one size fits all” approach, the knowledge gained from LCD helps us tailor the algorithmic improvements (perceptual-oriented or RL-oriented) for a specific game.

**Summary Of The Review:**

In summary, the paper conducts solid experiments to discover a novel taxonomy of challenges in the Procgen benchmark, though I have concerns on how such knowledges can benefit future research. I would like to hear the authors' opinions on it.

---

> ### Author Response · Authors · 2022-11-19
> **Response**
>
> > In Section 4, the authors choose 7 games for pretraining visual representations with self-supervision. I am wondering why these 7 games are chosen. In other words, I don't quite get what "consistent motion" means.
>
>
> Thanks for the question. By “consistent motion” we mean scenarios where the agent is moving versus scenarios where the agent is pinned to the center of the screen.
>
> > The results in Figure 3 seem to be dominated by two games (Leaper and Climber). For other games, the improvements look similar across easy and hard ϕ. Thus, I do not think the conclusion drawn from the results is convincing enough.
>
> We apologize for the confusion. In **Fig. 3a**, we show agent performance on the *training* levels as a function of the number of training steps. In **Fig. 3b**, we show agent *generalization* performance on novel levels from the most performant weights in **Fig. 3a**. We have added the table below to the manuscript (*SI A.7*), which indicates that in generalization the trends we described for easy versus hard ϕ are consistent across individual games.
>
> **easy $\phi$ % improvement per game** 	| 7.50%  	| -117.65% 	| 37.92%   	| -39.81% 	|
>
>  **hard $\phi$ % improvement per game** 	| 28.68% 	| 71.38    	| 3364.65% 	|
>
> > In Page 8 (Reinforcement learning challenge evaluations), the authors state that "the improvements in performance from reward shaping significantly correlated with the values of each game tested here". However, it depends on the reward shaping methods used (different envs use different reward shaping). So it looks to me that such correlation does not imply anything meaningful.
>
> Our taxonomy predicts that Leaper and Maze are challenging because of reward sparsity. Reward shaping here is only being used to validate that taxonomy. Regardless of the particular method, any reward shaping will alleviate reward sparsity. As our taxonomy predicts, reward shaping does help performance on Leaper and Maze, validating that these are RL “hard” games.
>
> > In "The whole is only sometimes greater than the sum of the parts in dRL" on Page 8, I do not fully understand the connection between the experiments setup (i) /(ii) and the implication of learning perception and action together/separately. Can the authors elaborate on it?
>
> We apologize for the lack of clarity. This experiment provides a third taxonomy of tasks according to whether improved perception and policy learning algorithms can interact synergistically. Indeed, as several reviewers have pointed out, perception and RL challenges may not be entirely independent, and this experiment identifies the tasks where they interact.
>
> To elaborate, we used PPO with pixel inputs as our baseline; this represents standard perception and a standard RL algorithm. In **Fig. 5**, we compare “(i) the performance of a PPG agent trained on semantic segmentation inputs minus baseline, and (ii) the performance of a PPG agent trained on pixels minus baseline plus the performance of a PPO agent trained on semantic segmentation inputs minus baseline (Fig. 5).” In other words, is the sum of learning with idealized perception and RL greater than the whole of these two parts? For the games in the Green triangle, the answer is yes, meaning if you are testing whether your algorithm can leverage synergies between perception and RL, these are the games to focus on.
>
> > In Figure 9, for some games, using p < 1 yields much better performance. Do the authors have any interpretation of such results?
>
> An excellent question. We believe that changing the stochasticity of reward feedback alters the likelihood of encountering different local optima for a small subset of games (though the global optimal policy is conserved) over different learning trajectories. The phenomenon pointed out by the reviewer is accounted for by our metrics in the LCD taxonomy (*SI A.4*).

---

> > ### Comment · Reviewer_SeAm · 2022-11-30
> > **Thanks for the reply!**
> >
> > Thank the authors for addressing my questions. However, some points are still not very clear.
> >
> > > Choice of the 7 games and results in Fig.3
> >
> > Can the authors elaborate more on why the self-supervised approach is not suitable for games where the agent is pinned to the center? Besides, in `Maze`, `Heist`, `Miner`, `Fruitbot` and `Plunder`, the agent is also not pinned to the center. Why exclude these environments? Since the results in Fig.3 are obtained from the same set of 7 games, a solid justification for the choice of games is important. My thinking is that for empirical works, we should not pick a subset of a common benchmark unless there are strong reasons to do so.
> >
> > > Correlation between performance improvements brought by reward shaping and the $\psi$ value of each game
> >
> > To me, the argument in the paper conveys the message that a lower $\psi$ value suggests less improvement from reward shaping. However, current experiments are only on games with high $\psi$. It is possible that games with lower $\psi$ also benefit from reward shaping. I suggest removing the correlation statement because it would send the wrong messages to readers.

---

### Official Review · Reviewer_tmph · 2022-10-24

**Confidence:** 4
**Correctness:** 2
**Technical Novelty And Significance:** 2
**Empirical Novelty And Significance:** 1
**Recommendation:** 3

**Clarity, Quality, Novelty And Reproducibility:**

This paper is well structured, but doesn't have enough  clarity and quality to support claims, especially the effectiveness of the proposed taxonomy.
The novelty is low.  This paper applies basic self-supervised representation learning  using some features often used in computer vision for the effectiveness of the proposed taxonomy.

**Strength And Weaknesses:**

The strength of this paper is to incorporate recent great successes of visual representation learning into deep reinforcement algorithms and to propose new taxonomy of challenges in the Procgen bechmark. The weakness of this paper is the definition of perception  challenge as the structure of the scenes and the definition of reinforcement learning as the total rewards. Those definitions are very  limited and evaluations on more variations of the definitions would support the claims.


**Summary Of The Paper:**

This paper introduces the Learning Challenge Diagnosticator (LCD) that is a tool to measure the contribution of advances I visual representation learning and the effectiveness of reinforcement learning algorithms at discovering policies to the successes of deep reinforcement learning models. The proposed approach is applied to challenges in Procgen benchmark and shows the effectiveness using  the self-supervised visual representations and  the reward shaping.

**Summary Of The Review:**

Incorporating the recent advances of visual representations learning is good, and
proposing the environment to measure the performance of perceptual part and reinforcement learning part is good. But the  definitions of levels of perception and
difficulties of reinforcement learning is not well designed and considered.

---

> ### Author Response · Authors · 2022-11-19
> **Response**
>
> > The weakness of this paper is the definition of perception challenge as the structure of the scenes and the definition of reinforcement learning as the total rewards. Those definitions are very limited and evaluations on more variations of the definitions would support the claims.
>
> Our LCD acts as a sensitivity analysis, where the perceptual or RL problem in a game is perturbed while making the other problem as structured and easy-to-learn as possible. As we wrote, the important decision is to “...begin by modifying a game to put its perceptual representations and reinforcement learning problem under experimental control.“ For the perceptual problem, “we tested the impact of learning on three different types of visual inputs, which varied in the amount of structure in the environment they conveyed to agents: the original frame, figure-ground segmentations of each frame, or semantic segmentations of each frame (Fig. 1b,c).” Our use of these three levels of perception also closely aligns with the levels of representation that are speculated to arise in human vision, and develop over the course of viewing a scene [1].
>
> For the RL problem, we considered reward sparsity because it has been found to degrade learning signal, making it difficult for RL agents to learn anything [2, 3, 4]. Arbitrarily changing the reward function is fraught with the risk of changing the task’s optimal policy. Our manipulation is designed to decrease the learning signal’s saliency without altering the optimal policy.
>
> We have expanded our discussion to expand on the philosophy behind perceptual/RL perturbations, and what motivated our choices for the experiments in this manuscript.
>
> > This paper is well structured, but doesn't have enough clarity and quality to support claims, especially the effectiveness of the proposed taxonomy.
>
> We have demonstrated our taxonomy’s effectiveness (**Fig 1d**) in several ways. In **Fig 3**, we validate our taxonomy of perceptually “hard” games. Our taxonomy correctly predicts which games benefit more from better perception. We also validate our taxonomy of “hard” RL games in Section 4. Our taxonomy also predicts which games benefit more from a stronger RL algorithm. In **Fig 4**, we show that our taxonomy generalizes across different RL algorithms. And in **Fig 5**, we provide another taxonomy of games according to synergistic benefits between perception and RL. In several ways, we have shown that our taxonomy is effective and useful.
>
> > The novelty is low.
>
> We provide the first (to our knowledge) systematic approach to diagnosing the computational challenges faced by dRL agents in video games. **This is important** because our results will allow researchers to identify the challenges they are facing in RL benchmarks like Procgen, focus their efforts and computational resources on those games that best capture those challenges, and more efficiently develop better RL algorithms. For instance, if one is trying to develop better RL perception, they should focus on only the three games we identify as perceptually hard, rather than all 16 Procgen games. That saves significant time, resources, and work for generating better RL.
>
> [1] Lamma V and Roelfsema P. 2000. The distinct modes of vision offered by feedforward and recurrent processing. Trends in Neurosciences.
>
> [2] Jaderberg M et al., 2017. Reinforcement learning with unsupervised auxiliary tasks. ICLR.
>
> [3] Houthooft R et al., 2016. VIME: Variational information maximizing exploration. NeurIPS.
>
> [4] Andrychowicz M et al., 2016. Hindsight experience replay. NeurIPS.

---

### Official Review · Reviewer_8UAX · 2022-10-24

**Confidence:** 3
**Correctness:** 3
**Technical Novelty And Significance:** 2
**Empirical Novelty And Significance:** 3
**Recommendation:** 6

**Clarity, Quality, Novelty And Reproducibility:**

The paper is well-written and novel. The appendix provides a detailed set of reward plots for all 16 games with different perceptual representations and reward sparsity. The work discusses how the taxonomy correlates with techniques needed to improve the training. The authors present the code for parameterized ProcGen environments, training, and computational details for reproducibility.

### Minor clarifications/suggestions:

Figure 3b is unclear in terms of what is the takeaway message.

**Strength And Weaknesses:**

### Strengths

The paper presents a task taxonomy separated by perceptual and reinforcement learning. This is a novel reward performance-based taxonomy, for which the authors train policies in different perceptual and reward representations in each environment.

The work also has a detailed appendix with performance plots for 16 games under different perturbations.

The paper discusses interesting effects of the assumption that Leaper does not benefit from the self-supervised pretraining to predict optic flow as the color attributes of the object are lost which are important for it to solve the task.

### Weaknesses

The paper motivates the introduction with DQN but does not consider any off-policy algorithm while discussing the reinforcement learning challenge or reward scheme. Why are only on-policy algorithms (like PPO, and PPG) considered?

While the motivation of ranking tasks based on perceptual and reinforcement challenges is clear, the choice of input representation in pixels, figure-background vs semantic segmentation, and reward schemes are limited and not fully motivated. Estimating the mean reward achieved by the agents could be possible due to some spurious correlations at times, which the current process does not take into account. The perceptual and reinforcement learning challenge scores are computed separately while holding the other part appears on the easiest possible level. An underlying assumption is that the two challenges can be completely disentangled, which may not always be true if other types of perturbations are considered.

The proposed taxonomy depends on the DRL algorithm (PPO) and semantic segmentation representation used and does not use any task-intrinsic elements directly. Is there any theoretical or empirical evidence on how much the taxonomy would shift when other DRL algorithms or perceptual representations are used to compute the respective scores?

While the authors present a paragraph in section 4 on how co-training perception for action helps in 6 out of 16 games, with 3 being on the separating line, the readers will benefit from discussing or cross-referencing this in section 3. The paper discusses a separate policy trained for each of the 16 games and does not discuss possible perceptual and task-solving benefits of learning one policy for all [Reed et al., 2022 [A Generalist Agent](https://arxiv.org/pdf/2205.06175.pdf)].

Finally, an assumption used here is “training DNNs to predict the optic flow between successive frames of video can induce the ability to segment object-like superpixels from complex scenes” from Liu et al. 2021. Does this only hold true when the camera is fixed, such that only objects move in computing the optic flow with a fixed background? Can this assumption be applied to an agent in First-Person-Shooter (FPS) videogames with egocentric perceptual inputs?


**Summary Of The Paper:**

The paper presents a taxonomy of the tasks, namely the Learning Challenge Diagnosticator (LCD), based on the perceptual and reinforcement learning challenges in the Procgen benchmark. The games are parameterized to perturb the perceptual representation (original pixels, figure-ground or semantic segmentations) or to vary the sparsity of the reward scheme. The integrated reward trajectories of a set of agents of sampled perturbed environments are used to compute scores that describe the perceptual and reinforcement challenge of that game. Specifically, the score is the normalized average absolute AUC difference of a policy across all changes; where a score of 0 indicates sparse reward is not much of a challenge while 1 represents a significant RL challenge. These agents are trained with either PPO or PPG for 200M steps (~24hrs per game). The main advantage of LCD is that it reveals failure cases and instructs algorithmic development. For the perceptual challenge, the work proposes self-supervised pretraining on in-domain videos to compute the optic flow between two frames giving a better visual representation for policy learning than the alignment (clip-ViT-B-32) model trained on images and captions. For the reinforcement challenge, the work demonstrates reward shaping on 3 games: Heist, Leaper, and Maze, demonstrating how the improvements correlated with the LCD taxonomy scores for RL complexity.

**Summary Of The Review:**

Overall, the paper presents an interesting study of how abstractness in perceptual representations and reward sparsity affects performance in video games and motivates the readers to broadly think about the issues in solving perceptual challenges with pretraining and reinforcement challenges with reward shaping. However, certain assumptions for pretraining and evaluation are not fully justified. A discussion in highlighting the potential limitations of these assumptions will clarify the takeaways of this work.

----
References:

A few relevant citations for visual pre-training: Xiao et al. “Masked Visual Pre-training for Motor Control.” *ArXix* abs/2203.06173 (2022)

---

> ### Author Response · Authors · 2022-11-19
> **Response 1/2**
>
> > Why are only on-policy algorithms (like PPO, and PPG) considered?
>
> We did not design the LCD with any specific RL algorithm in mind, and instead wanted to focus on the standards in the field and test how general findings from the LCD are. For that reason, we limited ourselves to rigorous experiments with the standard PPO and the more recent and performant PPG, which is state-of-the-art on Procgen [1,2]. Importantly, we show that findings derived from the LCD on agents trained with PPO generalize and are predictive for those trained with PPG.
>
> > While the motivation of ranking tasks based on perceptual and reinforcement challenges is clear, the choice of input representation in pixels, figure-background vs semantic segmentation, and reward schemes are limited and not fully motivated.
>
> Our LCD implements a sensitivity analysis, where the perceptual or RL problem in a game is perturbed while making the other problem as structured and easy-to-learn as possible. As we wrote, the important decision is to “...begin by modifying a game to put its perceptual representations and reinforcement learning problem under experimental control.“ For the perceptual problem, “we tested the impact of learning on three different types of visual inputs, which varied in the amount of structure in the environment they conveyed to agents: the original frame, figure-ground segmentations of each frame, or semantic segmentations of each frame (Fig. 1b,c).” Our use of these three levels of perception also closely aligns with the levels of representation that are speculated to arise in human vision, and develop over the course of viewing a scene [3].
>
> For the RL problem, we considered reward sparsity because it degrades the learning signal, making it difficult for RL agents to learn anything [4, 5, 6]. As we prove in the Supplemental Information *A.3*, our reward manipulation changes the difficulty of learning a policy while preserving a game’s optimal policy.
>
> We have expanded our manuscript to elaborate on the motivations for our choice of perceptual/RL perturbations.
>
> > Estimating the mean reward achieved by the agents could be possible due to some spurious correlations at times, which the current process does not take into account.
>
> If agents relied on visual shortcuts to learn Procgen games, they would have performed better with pixel inputs than semantic ones because semantic masks would disrupt these shortcuts. However, the opposite was true for all but one game (Leaper), for which semantic labels discarded information agents needed to solve the task (object colors). Spurious correlations and shortcuts did not affect our findings with the LCD.
>
> > The perceptual and reinforcement learning challenge scores are computed separately while holding the other part appears on the easiest possible level. An underlying assumption is that the two challenges can be completely disentangled, which may not always be true if other types of perturbations are considered.
>
> Our assumptions yielded a taxonomy (Fig. 1D) that we validated throughout the paper. Our findings indicate that this taxonomy should guide the development of RL agents. In the case of Procgen, focus only on Maze, Caveflyer, Dodgeball, Plunder, Coinrun, Ninja, Climber, and Starpilot when tuning the perceptual systems of dRL agents, and focus on Leaper, Bigfish, and Maze for tuning RL algorithms.
>
> > The proposed taxonomy depends on the DRL algorithm (PPO) and semantic segmentation representation used and does not use any task-intrinsic elements directly. Is there any theoretical or empirical evidence on how much the taxonomy would shift when other DRL algorithms or perceptual representations are used to compute the respective scores?
>
> We used the LCD to develop a taxonomy for PPO *and* PPG. PPO is a strong, popular, and efficient to compute baseline, whereas PPG is more performant but also more computationally expensive. The PPO taxonomy predicts the PPG taxonomy (**Fig. 4**), meaning that the taxonomies we provide in our paper generalize and can be used to guide development of future RL algorithms.
> Figure/ground and semantic segmentation representations are derived from the game engine itself, making them task-intrinsic. The same can be said for our RL perturbations to induce reward sparsity.
>
> > While the authors present a paragraph in section 4 on how co-training perception for action helps in 6 out of 16 games, with 3 being on the separating line, the readers will benefit from discussing or cross-referencing this in section 3.
>
> We have revised section 3 to forecast the results in section 4. Thanks for this suggestion.

---

> > ### Author Response · Authors · 2022-11-19
> > **Response 2/2**
> >
> >
> > > The paper discusses a separate policy trained for each of the 16 games and does not discuss possible perceptual and task-solving benefits of learning one policy for all [Reed et al., 2022 A Generalist Agent].
> >
> > This is a great point: it is possible that multi-game training could eventually ameliorate the perceptual or RL challenges that we identify here. However, in its current state, GATO, the algorithm that the reviewer mentioned, is nowhere near close to as performant as PPG trained on individual Procgen games (mean normalized scores: PPO 57.6%, GATO 60.8%, PPG 75.7%), and also requires industry-scale resources which we do not have access to [7]. We see a future where the algorithms of large-scale modeling efforts like GATO are tuned via LCD to more efficiently learn how to handle perceptual and RL challenges across many different video game suites.
> >
> > > Finally, an assumption used here is “training DNNs to predict the optic flow between successive frames of video can induce the ability to segment object-like superpixels from complex scenes” from Liu et al. 2021. Does this only hold true when the camera is fixed, such that only objects move in computing the optic flow with a fixed background? Can this assumption be applied to an agent in First-Person-Shooter (FPS) videogames with egocentric perceptual inputs?
> >
> > Very interesting question. Optic flow was originally proposed as a mechanism observers rely on to estimate their heading [8]. Optic flow algorithms, such as the one we rely on for our perceptual front-end, work in the first person view, with or without camera motion. We are in the process of verifying this result by extending our LCD and perceptual front-end to a novel game with a first-person view, Avalon [9], and will add these results as soon as they are finished.
> >
> > [1] Agarwal R et al., 2021. Deep Reinforcement Learning at the Edge of the Statistical Precipice. NeurIPS.
> >
> > [2] Cobbe et al., 2020. Phasic Policy Gradient.
> >
> > [3] Lamma V and Roelfsema P. 2000. The distinct modes of vision offered by feedforward and recurrent processing. Trends in Neurosciences.
> >
> > [4] Jaderberg M et al., 2017. Reinforcement learning with unsupervised auxiliary tasks. ICLR.
> >
> > [5] Houthooft R et al., 2016. VIME: Variational information maximizing exploration. NeurIPS.
> >
> > [6] Andrychowicz M et al., 2016. Hindsight experience replay. NeurIPS.
> >
> > [7] https://paperswithcode.com/sota/reinforcement-learning-on-procgen
> >
> > [8] Gibson, J. J. (1950). The Perception of the Visual World. Riverside, CA: Riverside Press.
> >
> > [9] Albrecht J et al., 2022. Avalon: A benchmark for RL generalization using procedurally generated worlds. NeurIPS.

---

> > > ### Comment · Reviewer_8UAX · 2022-12-03
> > > **Acknowledgement of the authors' rebuttal**
> > >
> > > I appreciate the authors' responses to my questions. I hope that the additional results on Avalon will be updated in the main paper.
> > > I will retain my previous score.

---

### Author Response · Authors · 2022-11-19
**Response to all reviewers**

Thank you to the reviewers for taking the time to read and review our paper. Your critiques are sharp and insightful, and we found that all four reviewers converged on the following two issues, which we address (or are in the process of addressing) through this rebuttal:

1. What does the Learning Challenge Diagnosticator (LCD) tell us about RL agents and challenges?
2. What is the novelty of the LCD?

We are working on addressing these issues in the following ways:

1. What does the LCD tell us (**UAX**, **tmph**, **SeAm**, **Ni8m**)?
**Response:** An open problem in the field of deep RL is to understand the computational challenges that agents face on different games. To what extent is a game difficult because of its perceptual versus reinforcement learning challenge? Answering this question will allow researchers to identify the challenges they are facing in RL benchmarks, focus their efforts and computational resources on those games that best capture those challenges, and more efficiently develop better RL algorithms.
The LCD is designed to address this problem and provide the field of RL with a first-of-its-kind diagnostic tool for evaluating the computational challenges of video games. The LCD’s predictions of hard vs. easy games for perception vs. RL improve sensitivity for designing algorithms that address these challenges (**Fig. 3** and **Section 4**).

2. Novelty and clarifications (**8UAX**, **Ni8m**).
**Response:** There was confusion about whether or not the point of this paper is to introduce representational learning and reward shaping into deep RL algorithms to improve performance on Procgen. We apologize for the confusion. To clarify: we introduced these algorithms to validate predictions from the LCD, and show its promise for guiding development of algorithms that can address the perceptual vs. RL challenges in video games like those in Procgen.

In addition to the above results, we have added citations and clarifications. We are also working on extending the LCD to another RL challenge, the recently released Avalon. We will update the manuscript with these experiments as soon as they are completed.

We have addressed each reviewer’s remaining comments directly. We thank the reviewers for their suggestions which have helped us refine our manuscript and bolstered the evidence for our findings.

---

### Decision · Program_Chairs · 2023-01-20

**Decision:**

Reject

**Justification For Why Not Higher Score:**

Arguably needs more work. Trying on different games, broadening the concept of perceptual difficulty.

**Justification For Why Not Lower Score:**

N/A

**Metareview: Summary, Strengths And Weaknesses:**

I would really want this paper to be in an acceptable state, as I find the topic fascinating and the idea novel. The problem addressed is how much of the challenge in an RL problem is in perceptual processing, and how much is in sparse rewards. In essence, the method is to vary both perceptual (graphical) elements and reward sparsity and see how it affects agents. I'm enthusiastic about the overall approach and would love to see it developed further. However, the paper as it stands arguably doesn't take it far enough. In particular, it only tests the method on seven games from ProcGen and the way it varies perceptual difficulty is rather limited. It could also be argued that we don't actually learn all that much from the experiments in the paper.